# Efficient Online Variational Estimation via Monte Carlo Sampling

**Mathis Chagneux** [1]   **Mathias Müller** [2]   **Pierre Gloaguen** [3]   **Sylvain Le Corff** [4]   **Jimmy Olsson** [2]

## Abstract

This article addresses online variational estimation in parametric state-space models. We propose a new procedure for efficiently computing the evidence lower bound and its gradient in a streaming-data setting, where observations arrive sequentially. The algorithm allows for the simultaneous training of the model parameters and the distribution of the latent states given the observations. It is based on i.i.d. Monte Carlo sampling, coupled with a well-chosen deep architecture, enabling both computational efficiency and flexibility. The performance of the method is illustrated on both synthetic data and real-world air-quality data. The proposed approach is theoretically motivated by the existence of an asymptotic contrast function and the ergodicity of the underlying Markov chain, and applies more generally to the computation of additive expectations under posterior distributions in state-space models.

## 1. Introduction

This work considers *state-space models* (SSMs) where the law of observations $(Y_t)_{t\in\mathbb{N}}$ is governed by an unobserved, or 'hidden', Markov chain $(X_t)_{t\in\mathbb{N}}$, and the finite dimensional distributions of $(X_t, Y_t)_{t\in\mathbb{N}}$ are given by parametric distributions indexed by $\theta \in \Theta$. Learning the parameter $\theta$ in this context is a complex task, as it usually requires access to the *joint-smoothing distributions*, defined at time $t$ as the conditional distribution of the latent states $X_{0:t}$[1] given the corresponding observations $Y_{0:t}$. This paper addresses the challenging problem of *online estimation* in SSMs, which

consists of sequentially learning both $\theta$ and the smoothing distributions as data stream in real time.

A first possible approach to this task is based on sequential Monte Carlo (SMC) smoothing (see Chopin & Papaspiliopoulos, 2020 and the references therein). Such SMC-based methods for online learning come with strong theoretical guaranties (Le Corff & Fort, 2013; Olsson & Westerborn, 2017; Tadić & Doucet, 2020; Gao et al., 2025), but typically suffer from the curse of dimensionality in the state dimension. This recently motivated the use of variational inference (VI) (Blei et al., 2017) as an alternative, as VI has demonstrated strong empirical performance for online inference in large-scale settings, particularly through stochastic variational frameworks (Hoffman et al., 2013; Broderick et al., 2013).

Taking a variational approach, the smoothing distributions are approximated by simpler distributions, the *variational distributions*, depending on some unknown parameter $\phi \in \Phi$. Both $\theta$ and $\phi$ are then learned by maximizing a proxy of the log likelihood, the *evidence lower bound* (ELBO), which is generally done using gradient ascent, thus requiring the computation of the ELBO's gradient. A key feature of these approaches is the choice of the variational distribution, which must be suited for online learning. Campbell et al. (2021) rely on structured variational distributions that both mimic the Markovian form of the smoothing distributions (see Section 3) and are well suited to online learning. The authors explore online variational additive smoothing for the recursive computation of the ELBO and its gradients, using a Bellman-type recursion inspired by reinforcement learning and recursive maximum likelihood methods. A major limitation of this algorithm is that, *at each iteration $t$*, it requires solving an inner optimization problem to learn a regression function serving as a proxy for a conditional expectation. This step can be computationally expensive and depends critically on the appropriate choice of the regression class. More broadly, a key drawback of existing online variational learning methods is the lack of theoretical guarantees for the proposed algorithms.

In this paper, we develop a theoretically grounded framework for online variational learning. Our approach is rooted in a stochastic approximation perspective, and, following Mastrototaro et al. (2025), we show that for a structured

[1]Télécom Paris, Institut Polytechnique de Paris, Palaiseau, France [2]KTH Royal Institute of Technology, Stockholm, Sweden [3]Université Bretagne Sud, UMR CNRS 6205, LMBA, F-56000 Vannes, France [4]Sorbonne Université, Université Paris Cité, CNRS, Laboratoire de Probabilités, Statistique et Modélisation, LPSM, F-75005 Paris, France. Correspondence to: Mathias Müller <matmul@kth.se>.

*Proceedings of the 43rd International Conference on Machine Learning*, Seoul, South Korea. PMLR 306, 2026. Copyright 2026 by the author(s).

[1]$a_{u:v}$ is a short-hand notation for $(a_u, \ldots, a_v)$.

variational distributions parameterized by flexible function classes (such as deep neural networks), online variational learning amounts to maximizing a lower bound (COLBO) on the limiting time-normalized asymptotic log likelihood, also known as the asymptotic contrast function. Building on this theoretical framework, we first formulate an ideal—but generally intractable—stochastic approximation algorithm that maximizes the COLBO, and then propose a Monte Carlo version of then same, using an efficient importance sampling approach that avoids any regression task and outperforms the algorithm of Campbell et al. (2021) in terms of computation time. The main contributions of this paper can be summarized as follows:

- We provide a theoretically grounded framework for online variational learning, showing that it can be viewed as Robbins—Monro algorithm.

- We propose a computationally efficient online estimator of the COLBO its gradient in the context of SSMs. In contrast to computationally intensive SMC or Markov chain Monte Carlo (MCMC) methods, our algorithm, which we refer to as *Recursive Monte Carlo Variational Inference* (RMCVI) to emphasize its iterative structure, relies on simple i.i.d. samples from the marginal variational distributions.

- The proposed algorithm is not limited to the online computation and optimization of the COLBO and can be directly adapted to compute more general expectations, training losses, or gradients under distributions that admit a Markovian structure.

- Experimentally, we demonstrate the performance of our estimator both on synthetic and real world data.

## 2. Related work

Our methodology draws on recent advances in smoothing methods for SSMs by (i) proposing a Monte Carlo approach for approximating conditional expectations and (ii) relying on a structured variational family.

*SMC for online learning in SSMs.* The original approaches to online smoothing is based on SMC methods. We refer the reader to (Douc et al., 2014, Section 11) for a presentation of the general concepts underlying SMC algorithms, and to Olsson & Westerborn (2017); Gloaguen et al. (2022); Dau & Chopin (2023) for more recent developments and applications of these methods to online smoothing. Theoretical guaranties for online optimization of the log-likelihood using SMC can be found in (Le Corff & Fort, 2013; Olsson & Westerborn, 2017; Tadić & Doucet, 2020; Gao et al., 2025).

*Offline VI for SSMs.* Early works on VI for SSMs primarily focus on offline estimation (Johnson et al., 2016; Krishnan

et al., 2017; Lin et al., 2018), *i.e.*, they require prior access to the entire observation sequence $Y_{0:t}$ in order to compute the gradients of the ELBO. These approaches rely on a *forward* factorization of the variational distribution, which is incompatible with online learning.

*Online VI for SSMs.* Marino et al. (2018); Zhao & Park (2020); Dowling et al. (2023) opts to trade smoothing for filtering by targeting the marginal distributions at each timestep with variational distributions that depend only on observations up to $t$. By designing a new variational family, Campbell et al. (2021) show that the ELBO and its gradient can be recursively expressed via conditional expectations, providing a natural framework for online learning. The authors approximate these conditional expectations at each time step using functional approximations.

*Online Variational SMC.* A recent approach to improving SMC for online learning is to adapt the particle proposal dynamics by optimizing a variational objective (Zhao et al., 2022; Mastrototaro & Olsson, 2024; Mastrototaro et al., 2025). In this framework, the variational method aims to approximate the locally optimal proposal while simultaneously learning the model parameters, rather than targeting the full joint smoothing distribution.

*Theoretical guaranties for VI.* In the offline setting, Chagneux et al. (2024) established the first theoretical results on error control in VI for SSM, building on the variational family of Campbell et al. (2021). Moreover, Mastrototaro et al. (2025) provided an online variational SMC framework that provably maximizes a well-defined asymptotic contrast function via stochastic approximation, which serves as the conceptual inspiration for our work.

## 3. Model and background

Consider an SSM $(X_t, Y_t)_{t \in \mathbb{N}}$, where $(X_t)_{t \in \mathbb{N}}$ is a discrete-time Markov chain on $\mathsf{X} := \mathbb{R}^{d_x}$. The distribution of $X_0$ has density $\chi$ with respect to the Lebesgue measure $\mu$ and for all $t \in \mathbb{N}$, the conditional distribution of $X_{t+1}$ given $X_{0:t}$ depends only on $X_t$ and has transition density $m_t^\theta(X_t, \cdot)$. In SSMs, it is assumed that the states of the Markov chain are only partially observed through an observation process $(Y_t)_{t \in \mathbb{N}}$ taking on values in $\mathsf{Y} := \mathbb{R}^{d_y}$. For every $t \in \mathbb{N}$, the observations $Y_{0:t}$ are assumed to be conditionally independent given $X_{0:t}$ and such that the conditional distribution of each $Y_s$, $s \in [\![0, t]\!]$, given $X_{0:t}$ depends only on the corresponding $X_t$ and has density $g_t^\theta(X_t, \cdot)$ with respect to the Lebesgue measure. The model is then defined, for all $t \in \mathbb{N}$, by the joint distributions

$$p_{0:t}^\theta(x_{0:t}, y_{0:t}) := \chi(x_0) g_0^\theta(x_0, y_0)$$
$$\times \prod_{s=1}^t m_s^\theta(x_{s-1}, x_s) g_s^\theta(x_s, y_s) \quad (1)$$

of the hidden states and the observations.

A classical learning task in SSMs is *state inference*, which consists of estimating the *joint-smoothing distribution*, *i.e.* the conditional distribution of $X_{0:t}$ given $Y_{0:t}$, given by

$$\pi_{0:t}^{\theta}(x_{0:t}) := \frac{p_{0:t}^{\theta}(x_{0:t}, y_{0:t})}{p_{0:t}^{\theta}(y_{0:t})},$$

where $p_{0:t}^{\theta}(y_{0:t}) := \int p_{0:t}^{\theta}(x_{0:t}, y_{0:t}) \, dx_{0:t}$ is the observed-data likelihood. The marginal of this joint distribution with respect to the state $x_t$ at time $t$ is known as the *filtering distribution* at time $t$, and its density with respect to the Lebesgue measure is denoted by $\pi_t^{\theta}$. It is straightforward to show that the density of the joint-smoothing distribution satisfies the so-called *backward decomposition*

$$\pi_{0:t}^{\theta}(x_{0:t}) = \pi_t^{\theta}(x_t) \prod_{s=1}^{t} b_{s-1|s}^{\theta}(x_s, x_{s-1}), \qquad (2)$$

where each *backward kernel*

$$b_{s-1|s}^{\theta}(x_s, x_{s-1}) \propto m_s^{\theta}(x_{s-1}, x_s) \pi_{s-1}^{\theta}(x_{s-1}) \qquad (3)$$

is the conditional probability density function of $X_{s-1}$ given $(X_s, Y_{0:s-1})$. The backward decomposition stems from the fact that the hidden process is still Markov when evolving conditionally on the observations, with time-inhomogeneous transition densities (3). However, since the filtering distributions are intractable, the backward kernels generally lack closed-form expressions.

In variational approaches, the smoothing distribution $\pi_{0:t}^{\theta}$ is approximated by selecting a candidate from a parametric family $\{q_{0:t}^{\phi}\}_{\phi \in \Phi}$, known as the *variational family*, where $\Phi$ is a parameter space. This parameter is learned jointly with $\theta$ by maximizing the ELBO

$$\mathcal{L}_t^{\theta, \phi} = \mathbb{E}_{q_{0:t}^{\phi}} \left[ \log \frac{p_{0:t}^{\theta}(X_{0:t}, Y_{0:t})}{q_{0:t}^{\phi}(X_{0:t})} \right], \qquad (4)$$

where $\mathbb{E}_{q_{0:t}^{\phi}}$ denotes expectation under $q_{0:t}^{\phi}$.

Needless to say, the form of the variational family is crucial in this approach. Motivated by the backward decomposition, some works impose structure on the variational family through a factorization of $q_{0:t}^{\phi}$. A variational counterpart of (2), introduced by Campbell et al. (2021), is given by

$$q_{0:t}^{\phi}(x_{0:t}) = q_t^{\phi}(x_t) \prod_{s=1}^{t} q_{s-1|s}^{\phi}(x_s, x_{s-1}), \qquad (5)$$

where $q_t^{\phi}$ (resp. $q_{s-1|s}^{\phi}(x_s, \cdot)$) are user-designed probability density functions, the variational kernels, whose parameters are learned from data.

A decisive advantage of this factorization is that it respects the true dependencies in (2). Moreover, Chagneux et al. (2024) established an upper bound on the error when expectations with respect to the smoothing distribution are approximated by expectations with respect to variational distributions that satisfy this backward factorization.

**Defining the variational distributions.** The variational inference framework requires the definition of the variational distributions involved in (5), *i.e.* the set of distributions $(q_t^{\phi}, q_{t|t+1}^{\phi})_{t \in \mathbb{N}}$. We here define a new variational family in a recursive manner, with shared parameters over time. This new design based on backward factorization is efficient in terms of online parameter learning (as the number of parameters does not grow with $t$) and creates a link between variational kernels that will ensure an efficient importance sampling procedure in Section 5.

Our online learning challenge requires that variational distributions (i) can be recursively defined using streaming data $(y_t)_{t \in \mathbb{N}}$, (ii) are related to each other to mimic the relation given by (3) between the backward kernel and the filtering distribution, (iii) are easy to sample from to perform Monte Carlo approximations. For this purpose, each $q_t^{\phi}$ is chosen as a parametric distribution belonging to the exponential family (in our experiments, the Gaussian family in X), defined by some parameter $\eta_t$ belonging to a parameter space $\mathcal{E}$. More precisely, we define intermediate quantities $(a_t)_{t \in \mathbb{N}}$ belonging to some user-defined space A, initialized at some arbitrary value $a_0 \in A$ and governed by a deterministic recursion $a_t = \mathcal{A}^{\phi}(a_{t-1}, y_t)$. Based on these quantities, we let, for each $t$, $\eta_t = f^{\phi}(a_t)$. Here the mappings $\mathcal{A}^{\phi}$ and $f^{\phi}$ are used-defined. This framework creates a link between variational filtering distributions, in the spirit of the filtering recursions in SSMs. The variational backward kernels are then defined from on the basis of this flow of distributions by setting, for all $t \in \mathbb{N}_{>0}$,

$$q_{t-1|t}^{\phi}(x_t, x_{t-1}) \propto q_{t-1}^{\phi}(x_{t-1}) \psi_t^{\phi}(x_{t-1}, x_t), \qquad (6)$$

where $(\psi_t^{\phi})_{t \geq 0}$ are potential functions on $X^2$ of form $\psi_t^{\phi}(x_{t-1}, x_t) = \exp(\langle \tilde{\eta}_t, T(x_{t-1}) \rangle)$, with $\tilde{\eta}_t = \tilde{f}^{\phi}(x_t)$ and $T(x_{t-1})$ being a natural parameter and a sufficient statistic, respectively, for the chosen exponential family. Eqn. (6) ensures that $q_{t-1|t}^{\phi}(x_t, \cdot)$ will be a probability density function with natural parameter $\eta_{t-1|t} = \eta_{t-1} + \tilde{\eta}_t$. In this convenient setting, the backward kernels $q_{t-1|t}^{\phi}$ can have arbitrarily complex dependencies on $x_t$, while their densities are derived analytically from the potentials. This enables straightforward Monte Carlo sampling procedures and direct computations of normalizing constants (which are required in our proposed algorithm, *e.g.*, in (16) below), while at the same time avoiding the reduction of our variational kernels to mere transformations or linearizations (*e.g.*, linear Gaussian kernels). In is important to note that the

parameters of functions $\mathcal{A}^\phi$, $f^\phi$, $\tilde{f}^\phi$ are shared across time, leading to an amortized framework. In our experiments, these functions are neural networks, and $\phi$ are their weights.

## 4. Online variational learning

**The asymptotic contrast function and the COLBO.** In the context of maximum likelihood estimation, the online learning of an unknown model parameter $\theta$ is known as recursive maximum likelihood (RML) (Le Gland & Mevel, 1997). RML focuses on maximizing the *asymptotic contrast function* $\lambda(\theta) := \lim_{t\to\infty} t^{-1} \log p_\theta(Y_{0:t})$ (a.s.), which serves as a foundational objective in this setting. If the data are generated by an SSM belonging to the parametric family of interest, characterized by a 'true' parameter $\theta^*$, then, under suitable identifiability conditions, the asymptotic contrast is maximised at $\theta^*$. Consequently, the maximum likelihood estimator (MLE) is strongly consistent in the sense that it converges almost surely to $\theta^*$ as $t$ tends to infinity.

Since the asymptotic contrast $\lambda(\theta)$ is intractable, we use a similar approach to that of Mastrototaro et al. (2025) and instead aim to maximise online, with respect to $(\theta, \phi)$, the *contrast lower bound* (COLBO) given by

$$\ell(\theta, \phi) := \lim_{t\to\infty} \frac{1}{t} \mathcal{L}_t^{\theta,\phi} \le \lambda(\theta) \quad \text{(a.s.)}. \quad (7)$$

**Stochastic approximation viewpoint.** Following standard RML ideas, online variational learning seek to maximize $\ell(\theta, \phi)$ by updating $(\theta, \phi)$ in the direction of its gradient. Because $\ell(\theta, \phi)$ is defined as a long-run time average, it is natural to pursue a stochastic approximation approach with the goal of solving $\nabla_{\theta,\phi} \ell(\theta, \phi) = 0$. Indeed, defining $\mathcal{G}_t^{\theta,\phi} := \nabla_{\theta,\phi} \mathcal{L}_t^{\theta,\phi} - \nabla_{\theta,\phi} \mathcal{L}_{t-1}^{\theta,\phi}$, we may write

$$\frac{1}{t} \nabla_{\theta,\phi} \mathcal{L}_t^{\theta,\phi} = \frac{1}{t} \sum_{s=1}^t \mathcal{G}_s^{\theta,\phi} + \frac{1}{t} \nabla_{\theta,\phi} \mathcal{L}_0^{\theta,\phi}. \quad (8)$$

Interpreted through the lens of ergodic theory, it is tempting to see the long term limit of the right-hand side of (8) as an expectation, allowing $\nabla_{\theta,\phi}\ell(\theta, \phi)$ to be expressed as a *mean field* (*i.e.*, the deterministic drift) that governs the long-run behavior of Robbins–Monro stochastic updates. In this idealized framework, it is natural to use the observed gradient increments to build a sequence $(\theta_t, \phi_t)_{t\in\mathbb{N}}$ leading to an ideal procedure summarized in Algorithm 1. However, since each term $\mathcal{G}_s^{\theta,\phi}$ depends on the whole historical record $Y_{0:s}$ and $(Y_t)_{t\in\mathbb{N}}$ is not a Markov process, the existence of the limit of (8) as $t$ tends to infinity is non-trivial. Actually, to the best of our knowledge, no theoretical results exist justifying the existence of the COLBO objective (7) and its gradient, both of which are necessary to place this learning procedure on firm theoretical ground. In the coming sections we provide theoretical results motivating this existence

for the variational family of Section 3 (Eqn. (5) and (6)). In particular, we show that the COLBO objective and its gradient can be justified via the law of large numbers, applied to a suitably constructed Markov chain. On the basis of this justification, Algorithm 1 can be motivated as a stochastic approximation scheme with state-dependent Markov noise.

**Recursive expression of the ELBO and its gradient.** In the following, we assume that we are given a sequence $(y_t)_{t\in\mathbb{N}}$ of observations, and leave the dependence on these implicit in the notation. Write $\ell_0^{\theta,\phi}(x_{-1}, x_0) := \log(\chi(x_0) g_0^\theta(x_0, y_0))$, $q_{-1|0}^\phi(x_0, x_{-1}) = 1$ and for $t \in \mathbb{N}_{>0}$,

$$\ell_t^{\theta,\phi}(x_{t-1}, x_t) := \log\left( \frac{m_t^\theta(x_{t-1}, x_t) g_t^\theta(x_t, y_t)}{q_{t-1|t}^\phi(x_t, x_{t-1})} \right), \quad (9)$$

which allows to rewrite the ELBO (4) as

$$\mathcal{L}_t^{\theta,\phi} = \mathbb{E}_{q_{0:t}^\phi} \left[ \sum_{s=0}^t \ell_s^{\theta,\phi}(x_{s-1}, x_s) - \log q_t^\phi(x_t) \right].$$

The following proposition provides recursive formulas for computing both the ELBO and its gradient. For brevity, we let $\mathbb{E}_{t-1|t}^{\phi,x_t}$ denote expectation under $q_{t-1|t}^\phi(x_t, \cdot)$.

**Proposition 4.1.** *For every $t \in \mathbb{N}$ and $(\theta, \phi) \in \Theta \times \Phi$, the ELBO and its gradient are given by*

$$\mathcal{L}_t^{\theta,\phi} = \mathbb{E}_{q_t^\phi} [h_t(X_t)] - \mathbb{E}_{q_t^\phi} \left[ \log q_t^\phi(X_t) \right],$$

$$\nabla_\phi \mathcal{L}_t^{\theta,\phi} = \mathbb{E}_{q_t^\phi} \left[ \nabla_\phi \log q_t^\phi(X_t)\, h_t(X_t) + u_t(X_t) \right],$$

$$\nabla_\theta \mathcal{L}_t^{\theta,\phi} = \mathbb{E}_{q_t^\phi} [v_t(X_t)],$$

*where the real-valued function $h_t$ on $\mathsf{X}$ and its gradients*

---

**Algorithm 1** Ideal algorithm (exact recursions)
***
1: **For each** $t \in \mathbb{N}$, at the arrival of $y_t$, using Prop. 4.1:
2:     Compute $h_t, u_t, v_t$.
3:     Compute $\nabla_\theta \mathcal{L}_t^{\theta_t,\phi_t}$ and $\nabla_\phi \mathcal{L}_t^{\theta_t,\phi_t}$.
4:     **Update:**

$$\phi_{t+1} \leftarrow \phi_t + \gamma_{t+1}^\phi \left( \nabla_\phi \mathcal{L}_t^{\theta_t,\phi_t} - \nabla_\phi \mathcal{L}_{t-1}^{\theta_t,\phi_t} \right),$$

$$\theta_{t+1} \leftarrow \theta_t + \gamma_{t+1}^\theta \left( \nabla_\theta \mathcal{L}_t^{\theta_t,\phi_t} - \nabla_\theta \mathcal{L}_{t-1}^{\theta_t,\phi_t} \right),$$

where $(\gamma_{t+1}^\theta, \gamma_{t+1}^\phi)$ are learning rates satisfying the usual Robbins Monro conditions.

---

$u_t := \nabla_\phi h_t$ *and* $v_t := \nabla_\theta h_t$ *satisfy the recursions*

$$h_t(x_t) = \mathbb{E}_{t-1|t}^{\phi,x_t}\left[h_{t-1}(X_{t-1}) + \ell_t^{\theta,\phi}(X_{t-1}, x_t)\right]$$

$$u_t(x_t) = \mathbb{E}_{t-1|t}^{\phi,x_t}\left[u_{t-1}(X_{t-1}) + \nabla_\phi \log q_{t-1|t}^\phi(x_t, X_{t-1})\right.$$
$$\left. \times \{h_{t-1}(x_{t-1}) + \ell_t^{\theta,\phi}(x_{t-1}, x_t)\}\right]$$

$$v_t(x_t) = \mathbb{E}_{t-1|t}^{\phi,x_t}\left[v_{t-1}(X_{t-1}) + \nabla_\theta \ell_t^{\theta,\phi}(X_{t-1}, x_t)\right],$$

*with* $h_0(x_0) = \ell_0^{\theta,\phi}(x_{-1}, x_0)$, $u_0(x_0) = 0$, *and* $v_0(x_0) = \nabla_\theta \ell_0^{\theta,\phi}(x_{-1}, x_0)$.

*Proof.* See Appendix B. □

Proposition 4.1 is of twofold interest. First, it provides a recursive scheme for the online computation of the ELBO, which will lead to a learning algorithm in Section 5. Second, it highlights the natural quantities to consider when studying the existence of the COLBO objective and its gradient.

**Existence of the COLBO.** From now on, we assume that the observed data is generated by some SSM $(X_t, Y_t)_{t\in\mathbb{N}}$, which does not necessarily belong to the parametric family considered in Section 3. Under this assumption, it is easy to see that also the process $(Z_t)_{t\in\mathbb{N}}$, where $Z_t := (X_t, Y_t, h_t, u_t, v_t, a_t)$, with $h_t$, $u_t$, and $v_t$ being the functions defined recursively in Proposition 4.1 and $a_t$ being the intermediate quantities used in the parameterization of $q_t^\phi$, is a Markov chain. The state space and Markov kernel of $(Z_t)_{t\in\mathbb{N}}$ are denoted by $(\mathsf{Z}, \mathcal{Z})$ and $T^{\theta,\phi}$, respectively (see Appendix C, Eqn. (34), for details). The Markov property follows from the assumed SSM dynamics of $(X_t, Y_t)_{t\in\mathbb{N}}$, along with the fact that the updates of Proposition 4.1 , as well as the update of $a_t$ from $a_{t-1}$, are performed recursively based on the current observation $Y_t$. Denote also by $S^\phi$ the Markov kernel of the marginal chain $(X_t, Y_t, a_t)_{t\in\mathbb{N}}$. We will establish the exponential forgetting of the extended chain $(Z_t)_{t\in\mathbb{N}}$ under the following assumptions.

**Assumption 4.2.** There exist $\pi \in \mathsf{M}_1(\mathcal{X} \otimes \mathcal{Y} \otimes \mathcal{A})$ and $\alpha \in (0,1)$ such that for every $t \in \mathbb{N}_{>0}$, $\phi \in \Phi$, and $(x,y,a) \in \mathsf{X} \times \mathsf{Y} \times \mathsf{A}$,

$$\|(S^\phi)^t(x,y,a) - \pi\|_{\mathsf{TV}} \leq \alpha^t.$$

Assumption 4.2 is discussed in Appendix C.2. The following assumptions are purely technical.

**Assumption 4.3.** There exists $c \in \mathbb{R}_{>0}$ such that for every $\phi \in \Phi$, $a_t \in \mathsf{A}$, and $(x_{s+1}, y_{s+1}) \in \mathsf{X} \times \mathsf{Y}$,

(i) $\mathbb{E}_{s|s+1}^{\phi,x_{s+1}}\left[|\nabla_\phi \log q_{s|s+1}^\phi(x_{s+1}, X_s)|^2\right] \leq c^2$,

(ii) $\mathbb{E}_{s|s+1}^{\phi,x_{s+1}}\left[|\ell_s^{\theta,\phi}(X_s, x_{s+1})|^2\right] \leq c^2$,

(iii) $\mathbb{E}_{s|s+1}^{\phi,x_{s+1}}\left[|\nabla_\theta \ell_s^{\theta,\phi}(X_s, x_{s+1})|\right] \leq c$.

**Assumption 4.4.** There exist constants $0 < \varepsilon^- < \varepsilon^+ < \infty$ such that, for every $t \in \mathbb{N}_{>0}$, $(x_{t-1}, x_t) \in \mathsf{X}^2$, and $\phi \in \Phi$,

$$\varepsilon^- \leq \psi_t^\phi(x_{t-1}, x_t) \leq \varepsilon^+.$$

The following theorem establishes the geometric ergodicity of the extended Markov chain, if not for all $f$ in the space $\mathsf{F_b}(\mathcal{Z})$ of bounded measurable functions on Z, so at least for a subclass $\mathcal{L}(\mathcal{Z}) \subset \mathsf{F_b}(\mathcal{Z})$ of Lipschitz functions. More precisely, $f \in \mathcal{L}(\mathcal{Z})$ if there exists $\varphi \in \mathsf{F_b}(\mathcal{X} \otimes \mathcal{Y} \otimes \mathcal{A})$ such that for every $(x,y,a,h,h',u,u',v,v') \in \mathsf{X} \times \mathsf{Y} \times \mathsf{A} \times \mathsf{F_b}(\mathcal{X})^6$, writing $\mathsf{s} = (x,y,a)$,

(i) $|f(\mathsf{s}, h, u, v)| \leq \varphi(\mathsf{s})$,

(ii) $|f(\mathsf{s}, h, u, v) - f(\mathsf{s}, h', u', v')|$
$\leq \varphi(\mathsf{s})\left(\mathrm{osc}(h - h') + \mathrm{osc}(u - u') + \mathrm{osc}(v - v')\right)$,

where $\mathrm{osc}(f) = \sup_{z\in\mathsf{Z}} f(z) - \inf_{z\in\mathsf{Z}} f(z)$.

**Theorem 4.5** (Geometric ergodicity of $(Z_t)_{t\in\mathbb{N}}$)**.** *Assume 4.2–4.4. Then there exist* $\rho \in (0,1)$ *and a functional* $\zeta : \mathsf{F_b}(\mathcal{X})^6 \to \mathbb{R}_{>0}$ *such that for every* $(\theta,\phi) \in \Theta \times \Phi$, $t \in \mathbb{N}$, $f \in \mathcal{L}(\mathcal{Z})$, $z = (x_0, y_0, a_0, h_0, u_0, v_0) \in \mathsf{Z}$, *and* $z' = (x'_0, y'_0, a'_0, h'_0, u'_0, v'_0) \in \mathsf{Z}$,

$$|(T^{\theta,\phi})^t f(z) - (T^{\theta,\phi})^t f(z')|$$
$$\leq \zeta(h_0, h'_0, u_0, u'_0, v_0, v'_0)\|\varphi\|_\infty \rho^t. \quad (10)$$

*Moreover, there exists a functional* $\bar{\zeta} : \mathsf{F_b}(\mathcal{X})^3 \to \mathbb{R}_{>0}$ *and a kernel* $\Pi^{\theta,\phi}$ *such that for every* $f \in \mathcal{L}(\mathcal{Z})$, $\Pi^{\theta,\phi} f$ *is constant and for every* $t \in \mathbb{N}$ *and* $z = (x_0, y_0, a_0, h_0, u_0, v_0) \in \mathsf{Z}$,

$$|(T^{\theta,\phi})^t f(z) - \Pi^{\theta,\phi} f| \leq \bar{\zeta}(h_0, u_0, v_0)\|\varphi\|_\infty \rho^t. \quad (11)$$

*Proof.* See Appendix C. □

Although the contraction (11) does not hold in the total variation norm (due to the restriction to test functions in $\mathcal{L}(\mathcal{Z})$), the quantity $\Pi^{\theta,\phi}$ provided by the same theorem can be regarded as a candidate for the unique stationary distribution of $(Z_t)_{t\in\mathbb{N}}$. Moreover, by Proposition 4.1, each term $\mathcal{G}_s^{\theta,\phi} = \mathcal{G}^{\theta,\phi}\langle Z_{s-1:s}\rangle$ depends explicitly on the consecutive states $Z_{s-1:s}$ of the extended chain. By the law of large numbers for Markov chains, we may expect that (a.s.),

$$\lim_{t\to\infty} \frac{1}{t}\nabla_{\theta,\phi}\mathcal{L}_t^{\theta,\phi} = \iint \mathcal{G}^{\theta,\phi}\langle z, z'\rangle \Pi^{\theta,\phi}(\mathrm{d}z) T^{\theta,\phi}(z, \mathrm{d}z'). \quad (12)$$

Letting the limit (12) serve as the mean field of a stochastic approximation scheme with state-dependent Markov noise

(see Karimi et al., 2019b, Case 2), a recursive Robbins–Monro algorithm finding a stationary point of the COLBO gradient is given by

$$(\theta_{t+1}, \phi_{t+1}) \leftarrow (\theta_t, \phi_t) + \gamma_{t+1} \mathcal{G}^{\theta_t, \phi_t} \langle Z_{t-1}, Z_t \rangle,$$

and $Z_{t+1} \sim T^{\theta_{t+1}, \phi_{t+1}}(Z_t, \cdot)$, where $(\gamma_t)_{t \in \mathbb{N}_{>0}}$ is a sequence of step sizes satisfying the usual assumptions. This procedure is summarized in Algorithm 1, which uses distinct step-size sequences $(\gamma_t^\theta)_{t \in \mathbb{N}_{>0}}$ and $(\gamma_t^\phi)_{t \in \mathbb{N}_{>0}}$ for updating the model and variational parameters.

## 5. Online Monte Carlo approximation

We now derive a practical version of the ideal Algorithm 1. Proposition 4.1 suggests that it is possible to estimate the ELBO and its gradient recursively. The key feature of our Monte Carlo algorithm is that each conditional expectation in the recursion only needs to be estimated on a *finite support*, bypassing the regression step required at each time step in Campbell et al. (2021). This results in a more efficient procedure, as confirmed empirically in Section 6.2. The algorithm proceeds as follows:

First, sample $\{\xi_0^i\}_{i=1}^N \overset{\text{i.i.d.}}{\sim} q_0^\phi$, and set

$$\hat{h}_0^{\phi,i} = h_0(\xi_0^i), \quad \hat{u}_0^{\phi,i} = u_0(\xi_0^i), \quad \hat{v}_0^{\theta,i} = v_0(\xi_0^i).$$

At time $t \in \mathbb{N}_{>0}$, having access to a Monte Carlo sample $\{\xi_{t-1}^i\}_{1=i}^N$ from $q_{t-1}^\phi$ and approximations $\hat{h}_{t-1}^{\phi,i}, \hat{u}_{t-1}^{\phi,i}, \hat{v}_{t-1}^{\theta,i}$ of $h_{t-1}(\xi_{t-1}^i), u_{t-1}(\xi_{t-1}^i), v_{t-1}(\xi_{t-1}^i)$, respectively, sample independently $\{\xi_t^i\}_{i=1}^N$ from $q_t^\phi$ and update

$$\hat{h}_t^{\phi,i} = \sum_{j=1}^N \bar{w}_{t-1|t}^{\phi,i,j} \left( \hat{h}_{t-1}^{\phi,j} + \ell_t^{\theta,\phi}(\xi_{t-1}^j, \xi_t^i) \right), \qquad (13)$$

$$\hat{u}_t^{\phi,i} = \sum_{i=1}^N \bar{w}_{t-1|t}^{\phi,i,j} \left\{ \hat{u}_{t-1}^{\phi,j} + \nabla_\phi \log q_{t-1|t}^\phi(\xi_t^i, \xi_{t-1}^j) \right.$$
$$\left. \times \left( \hat{h}_{t-1}^{\phi,j} + \ell_t^{\theta,\phi}(\xi_{t-1}^j, \xi_t^i) \right) \right\}, \quad (14)$$

$$\hat{v}_t^{\theta,i} = \sum_{i=1}^N \bar{w}_{t-1|t}^{\phi,i,j} \left( \hat{v}_{t-1}^{\theta,j} + \nabla_\theta \ell_t^{\theta,\phi}(\xi_{t-1}^j, \xi_t^i) \right), \quad (15)$$

where

$$\bar{w}_{t-1|t}^{\phi,i,j} := \frac{q_{t-1|t}^\phi(\xi_t^i, \xi_{t-1}^j)/q_{t-1}^\phi(\xi_{t-1}^j)}{\sum_{k=1}^N q_{t-1|t}^\phi(\xi_t^i, \xi_{t-1}^k)/q_{t-1}^\phi(\xi_{t-1}^k)}. \quad (16)$$

Estimators (13–15) are *self-normalized importance sampling* estimators of the updates of Proposition 4.1, and (16) provides the (shared) importance weights of these estimators. Note that we cannot perform direct Monte Carlo approximation on the basis of samples from $q_{t-1|t}^\phi(\xi_t^i, \cdot)$, as we would not have access to any approximations of the values of the

functionals $h_{t-1}$ and $u_{t-1}$ at the sampled points. The use of importance sampling is therefore a prerequisite for updating the approximations. Moreover, note that the design of variational distributions imposed by (6) creates a link between the the target $q_{t-1|t}^\phi$ distribution and $q_t^\phi$, making the latter a natural proposal distribution.

Once $\{(\hat{h}_t^{\phi,i}, \hat{u}_t^{\phi,i}, \hat{v}_t^{\theta,i})\}_{i=1}^N$ are computed, approximations of the ELBO and its gradient at time $t$ are obtained by

$$\widehat{\mathcal{L}}_t^{\theta,\phi} = \frac{1}{N} \sum_{i=1}^N \left( \hat{h}_t^{\phi,i} - \log q_t^\phi(\xi_t^i) \right), \qquad (17)$$

$$\widehat{\nabla}_\phi \mathcal{L}_t^{\theta,\phi} = \frac{1}{N} \sum_{i=1}^N \left( \nabla_\phi \log q_t^\phi(\xi_t^i) \hat{h}_t^{\phi,i} + \hat{u}_t^{\phi,i} \right), \quad (18)$$

$$\widehat{\nabla}_\theta \mathcal{L}_t^{\theta,\phi} = \frac{1}{N} \sum_{i=1}^N \hat{v}_t^{\theta,i}. \qquad (19)$$

The full procedure, which we refer to as RMCVI (Recursive Monte Carlo Variational Inference), is detailed in Algorithm 2 in Appendix A. Appendix D provides refinements to increase computational efficiency and reduce the variance of the gradient estimator. It is worth noting that RMCVI computes estimated gradients using both $(\phi_t, \theta_t)$ and $(\phi_{t-1}, \theta_{t-1})$ (Eqns. (24–25), Appendix A), which introduces a deviation from the ideal updates of Algorithm 1. Such approximations—commonly employed in RML settings—are crucial for enabling a feasible practical online implementation.

**Impact of the bias.** The proposed algorithm introduces some approximations that are not covered by our theoretical analysis. A key difficulty in analyzing the practical implementation arises from the use of SNIS for the recursive gradients, as it introduces a bias of order $O(1/N)$. Recent literature provides several useful insights into this issue.

- A classical bias analysis could be carried out by adapting the analysis of Olsson & Westerborn (2017) to our Monte Carlo–based updates. This would allow us to control the discrepancy between the theoretical update and its empirical counterpart, thereby providing explicit finite-sample error bounds. In addition, it is known that this bias can be reduced using approaches such as those proposed in Cardoso et al. (2022). This suggests that the bias at each iteration can be controlled with an explicit dependence on $N$, meaning that the Monte Carlo sample size can be tuned across iterations to satisfy suitable conditions for convergence.

- Because we formulate our variational approach within a stochastic approximation framework, we can leverage recent results such as Karimi et al. (2019a) as well as more recent extensions (*e.g.*, Surendran et al., 2024).

These results make it possible to establish convergence to a critical point of the objective, with rates that explicitly depend on the bias at each iteration. This, of course, requires assumptions on the objective function, such as the Polyak–Łojasiewicz condition, $L$-smoothness, and assumptions on the mean-field bias, which in our setting is expected to scale as $1/N$.

# 6. Experiments

We now evaluate the proposed algorithm on several streaming data inference tasks.[2] Our goal is to demonstrate that the method can jointly learn both the latent posterior approximation and the model parameters in a fully sequential manner while requiring substantially less computation than existing online approaches (Table 1). Effective sample size diagnostics confirming the stability of the importance-sampling procedure across experiments are reported in Appendix E.4.

In all experiments, the variational filtering distributions $q_t^\phi$ are chosen to be in the Gaussian family. For the non-linear models (Sections 6.2 and 6.3) we implement the deterministic recursion $a_t = \mathcal{A}^\phi(a_{t-1}, y_t)$ as an RNN, where both the update function $\mathcal{A}^\phi$ and the parameter mapping $f^\phi$ are parameterized by MLPs with tanh activation functions. The variational backward kernels are defined via the potentials $\psi_t^\phi$ in (6), which are parameterized by similar neural networks (except for the linear Gaussian case, where we exploit analytical conjugation to derive exact backward kernels). In addition to this architecture, some control-variate tricks are implemented to reduce the variance of the gradient estimator (see details in Appendices D and E).

## 6.1. Linear-Gaussian HMM

We first assess our algorithm on a linear Gaussian SSM. Consider $(X_t, Y_t)_{t\in\mathbb{N}}$ in $\mathbb{R}^{d_x} \times \mathbb{R}^{d_y}$, with $X_0 \sim \mathcal{N}(\mu_0, Q_0)$, $Y_0 = GX_0 + \varepsilon_0$, and, for all $t \in \mathbb{N}_{>0}$,

$$X_t = FX_{t-1} + \nu_t, \qquad \nu_t \overset{\text{i.i.d}}{\sim} \mathcal{N}(0, I_{d_x}),$$
$$Y_t = GX_t + \varepsilon_t, \qquad \varepsilon_t \overset{\text{i.i.d}}{\sim} \mathcal{N}(0, I_{d_y}),$$

where $\theta = (F, G)$ is the parameter to be learned. In this case, the true smoothing distributions are Gaussian and can be computed via closed-form recursions (the Kalman smoother), providing an analytical reference for evaluation. The variational family $q_{0:t}^\phi$ is parameterized using Gaussian conditionals and marginals; see Appendix E.1 for details.

The first experiment is run with $d_x = d_y = 10$, and a sequence of $T = 50000$ observations. Figure 1 shows the evolution of the ELBO through the learning of $(\theta, \phi)$ as well as the posterior mean of the hidden states on a test sequence

[2]Code available at https://github.com/matmulKTH/online-variational-mc.

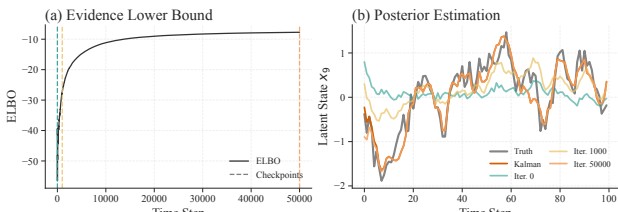

*Figure 1.* (a) Evolution of $\widehat{\mathcal{L}}_t^{\theta,\phi}/t$ during the online learning in the linear Gaussian SSM. Vertical lines indicate times at which state estimation is made on a test sequence. (b) State estimation on a test sequence (only one particular dimension is displayed).

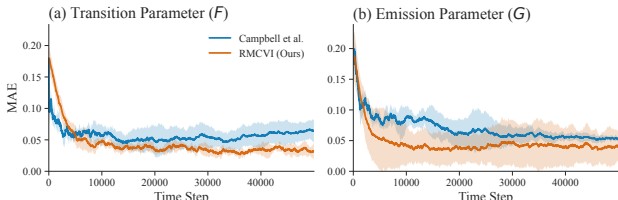

*Figure 2.* Model-parameter learning in the linear–Gaussian HMM. Mean-absolute errors of the transition ($F$) and emission ($G$) matrices for our method and Campbell et al. (2021).

of observations never seen by the model. The right panel shows a particular dimension of the hidden state for the test sequence, as well as the learned posterior distribution at different iterations. As the number of observations grows, our estimator learns a mapping that produces the true posterior distribution, corresponding to the oracle Kalman smoother. Figure 2 shows the MAE in parameter estimation of $F$ (a) and $G$ (b). Our method achieves comparable error than the regression based method of Campbell et al. (2021) approximately four times faster in our setup when using $N = 1000$ samples for the importance weights. Overall, these results highlight the computational efficiency of the proposed updates without compromising statistical accuracy.

**Empirical tightness of the COLBO** Since the COLBO (7) is a time-normalized limit of the ELBO, the gap to the asymptotic contrast $\lambda(\theta)$ is the limiting KL between variational and true smoothing distributions. We illustrate this on a 2D linear-Gaussian SSM, where $\lambda(\theta)$ and $\log p_\theta(Y_{0:t})$ are available in closed form via the Kalman filter. Figure 3 compares, on four independent datasets, our online COLBO estimate against $t^{-1} \log p_{\hat{\theta}_t}(Y_{0:t})$ and $t^{-1} \log p_{\theta^\star}(Y_{0:t})$. The learned $\hat{\theta}_t$ quickly matches the log-likelihood of $\theta^\star$, and the gap to our contrast vanishes as $\hat{\phi}_t$ learns the posterior.

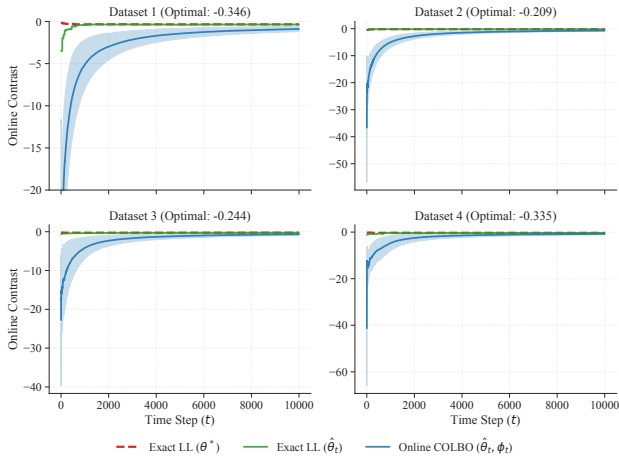

*Figure 3.* COLBO tightness on a 2D linear-Gaussian SSM. Each panel corresponds to an independent dataset; curves are averaged over 5 independent runs of our algorithm on that dataset (shaded: $\pm 1$ std).

## 6.2. Chaotic recurrent neural network.

We now consider the model used in Campbell et al. (2021), where $X_0 \sim \mathcal{N}(0, Q)$, $Y_0 = X_0 + \varepsilon_0$, and, for $t \in \mathbb{N}_{>0}$,

$$X_t = X_{t-1} + \frac{\Delta}{\rho}\left(\gamma W \tanh\left(X_{t-1}\right) - X_{t-1}\right) + \eta_t,$$

$$Y_t = X_t + \varepsilon_t,$$

where $(\eta_t)_{t \in \mathbb{N}_{>0}}$ and $(\varepsilon_t)_{t \in \mathbb{N}}$ are mutually independent sequences of i.i.d. $\mathcal{N}(0, Q)$ and Student-$t$ random variables, respectively. We set $d_x = d_y = 5$, and the use same true parameters as Campbell et al. (2021) (see Appendix E.2) and $N = 500$ importance samples.

**Filtering and one-step smoothing** To compare our method with the one of Campbell et al. (2021) on this model, we reproduce the 1-step smoothing experiment of their work (Campbell et al., 2021, Appendix B.2). Specifically, we evaluate the ability of both approaches to estimate the conditional laws of $X_{t-1}$ given $Y_{0:t}$ and of $X_t$ given $Y_{0:t}$ by learning $\mathbb{E}_{q^{\phi}_{t-1:t}}[X_{t-1}]$ and $\mathbb{E}_{q^{\phi}_t}[X_t]$. In order to perform the same comparison, we mimic the non-amortized framework of the original paper (details are provided Appendix E.2). Table 1 reports the 1-step smoothing and filtering errors as defined in Eqn. (61). We also report average computation time per gradient steps. With comparable errors, RMCVI is about 5 times faster than the regression approach.

**Online learning.** Moving beyond the fixed-parameter setting, we evaluate our method in a true streaming regime where both the variational parameters and selected generative parameters are learned online. Concretely, $T = 5 \times 10^5$ observations are processed with updates of the parameters $\gamma$ and $\rho$. Figure 4 (a) shows the MAE between parameter

estimates and the true value. The central panel shows the dynamics of $\widehat{\mathcal{L}}_t^{\theta,\phi}/t$ with checkpoint markers, at which state estimation on held-out sequences is performed (right panel) for a specific state dimension. These results show that the proposed scheme remains stable and accurate while simultaneously learning $(\rho, \gamma)$ in this highly nonlinear regime.

## 6.3. Air-quality Data

We evaluate the framework on the UCI Air-Quality dataset (Vito, 2008), which consists of hourly averaged responses from a chemical sensor array alongside meteorological data. We process the data into an 8-dimensional observation vector $y_t \in \mathbb{R}^8$ spanning approximately one year ($T \approx 9,300$ steps). This benchmark is characterized by frequent periods of sensor failure, where valid signals are absent for extended durations. Rather than imputing these irregularities offline, we process the stream directly to rigorously test the method's ability to maintain coherent belief states during blackout periods. Concretely, at missing time steps we omit the emission term $\log p_t^{\theta}(y_t \mid x_t)$ from the per-step ELBO contribution; the forward pass propagates dynamics through the gap, while the explicit backward kernels (Eqn. 5) enable smoothing-based reconstruction during blackouts. Visualizations of the data and more details on the signal characteristics and preprocessing are provided in Appendix E.3.

We model the air quality dynamics using a non-linear Gaussian SSM with residual transitions. Let $X_t \in \mathbb{R}^5$ and $Y_t \in \mathbb{R}^8$ denote the latent state and observations respectively. The generative process is defined as:

$$X_t = X_{t-1} + f_\theta(X_{t-1}) + \nu_t, \qquad \nu_t \sim \mathcal{N}(0, Q_\theta)$$
$$Y_t = g_\theta(X_t) + \varepsilon_t, \qquad \varepsilon_t \sim \mathcal{N}(0, R_\theta)$$

where $f_\theta$ and $g_\theta$ are neural networks parameterized by $\theta$ (with tanh activations), and the noise terms have diagonal covariance matrices, which are also learned. Here we used $N = 20$ importance samples for the Monte Carlo estimates.

**Online predictive performance.** Our primary focus is the model's performance in an online setting where all parameters must be learned from a cold start. We evaluate the model's ability to learn complex dynamics by measuring the one-step-ahead prediction RMSE on the five primary pollutants (CO, NO$_x$, NO$_2$, C$_6$H$_6$, O$_3$). We compare our RMCVI method against two baselines: a probabilistic online LSTM (Salinas et al., 2020) with a Gaussian output

| Method | 1-Smooth. | Filt. | Time |
|---|---|---|---|
| RMCVI (ours) | 8.9 (0.2) | 10.3 (0.2) | 1 ms |
| Campbell et al. (2021) | 9.2 (0.2) | 10.3 (0.2) | 4.8 ms |

*Table 1.* Time per gradient step, 1-step smoothing and filtering RMSE ($\times 10^{-2}$) (defined in Eqn. (61)) for the chaotic RNN.

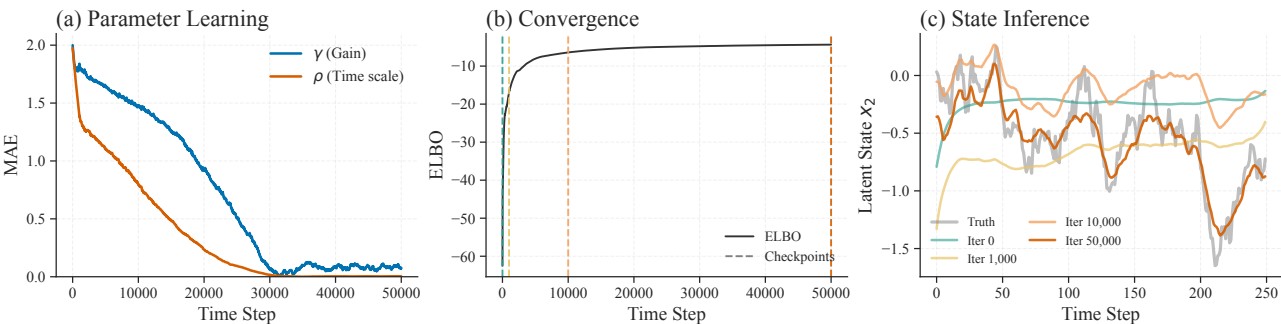

*Figure 4.* (a) Parameter MAE. (b) Approximate $\widetilde{\mathcal{L}}_t/t$ with checkpoint markers. (c) State estimation on a test sequence for one latent dimension. Colored lines/markers correspond to the same checkpoints.

trained sequentially via maximum likelihood, and Online Variational SMC (OVSMC, Mastrototaro & Olsson, 2024).

Figure 5 demonstrates the robustness and accuracy of our approach. As shown in Panel (a), we introduce a smoothing experiment with artificial sensor failure. While the purely autoregressive LSTM is limited to filtering and thus tracks the corrupted signal, RMCVI effectively recovers the underlying ground truth. Similarly, panel (b) confirms that RMCVI matches the predictive performance of the LSTM and outperforms OVSMC. Thus, RMCVI combines the forecasting power of autoregressive networks with the advantage of variational smoothing, all while maintaining lower computational costs than particle-based methods.

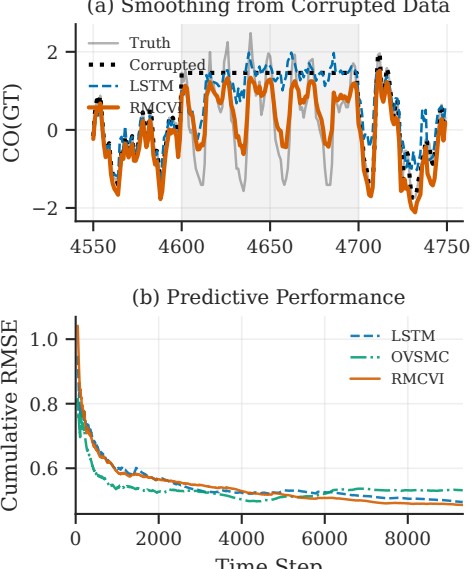

*Figure 5.* **Simultaneous Variational Learning and Prediction.** (a) Smoothing reconstruction during sensor failure (black dotted). RMCVI (orange) recovers the Truth (gray) while LSTM (blue) overfits the corruption. (b) Cumulative RMSE for one-step-ahead prediction averaged across all pollution features.

## 7. Discussion

We introduced a theoretically grounded online variational learning algorithm for SSM. The performance of our method is assessed with synthetic and real-world datasets, where it is shown to be more efficient than recent alternatives both for smoothing and prediction tasks.

Since the exact COLBO is a time-normalized limit of the ELBO, it has, in principle, the same tightness properties as the ELBO. Its tightness is governed by the expressiveness of the variational family, as in standard variational inference. Interestingly, the objective admits a natural extension, analogous to the importance-weighted autoencoder (IWAE), in which multiple samples from the variational distribution are used to form a tighter bound. This would yield a family of increasingly tight objectives that approach the true likelihood as increases. We leave a full development of this extension to future work. Another important future work concerns the theoretical analysis of the online Monte Carlo version of the algorithm, in particular of the bias of the Monte Carlo update, which is crucial to obtain quantitative guarantees and optimize hyperparameters.

## Acknowledgements

This work is supported by the Wallenberg AI, Autonomous Systems and Software Program (WASP), project "Online learning in dynamical generative models".

## Impact Statement

This paper presents work whose goal is to advance the field of Machine Learning. There are many potential societal consequences of our work, none of which we feel must be specifically highlighted here.

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

## A. Full online gradient estimator

---

**Algorithm 2** Online gradient estimator

---

1: **Input:** initial parameter estimate $(\theta_0, \phi_0)$, step sizes $\{\gamma_t^\theta, \gamma_t^\phi\}_{t \geq 1}$.
2: Sample $\{\xi_0^i\}_{i=1}^N$ independently from $q_0^{\phi_0}$;
3: For $1 \leqslant i \leqslant N$, set $\hat{h}_0^{\phi_0,i} \leftarrow h_0(\xi_0^i), \hat{u}_0^{\phi_0,i} \leftarrow 0, \hat{v}_0^{\phi_0,i} \leftarrow v_0(\xi_0^i)$.
4: Set $\widehat{\nabla}_\phi \mathcal{L}_0^{\theta_0,\phi_0} \leftarrow N^{-1} \sum_{i=1}^N \nabla \log q_0^{\phi_0}(\xi_0^i) \hat{h}_0^{\phi_0,i}$ and $\widehat{\nabla}_\theta \mathcal{L}_0^{\theta_0,\phi_0} \leftarrow N^{-1} \sum_{i=1}^N \hat{v}_0^{\phi_0,i}$.
5: Update $\phi_1 \leftarrow \phi_0 + \gamma_1^\phi \widehat{\nabla}_\phi \mathcal{L}_0^{\theta_0,\phi_0}$ and $\theta_1 \leftarrow \theta_0 + \gamma_1^\theta \widehat{\nabla}_\theta \mathcal{L}_0^{\theta_0,\phi_0}/$
6: **for** $t \geq 1$ **do**
7:   Sample $\{\xi_t^i\}_{i=1}^N$ independently from $q_t^{\phi_t}$.
8:   Set

$$\bar{w}_{t-1|t}^{\phi_t,i,j} \leftarrow \frac{q_{t-1|t}^{\phi_t}(\xi_t^i, \xi_{t-1}^j)/q_{t-1}^{\phi_{t-1}}(\xi_{t-1}^j)}{\sum_{k=1}^N q_{t-1|t}^{\phi_t}(\xi_t^i, \xi_{t-1}^k)/q_{t-1}^{\phi_{t-1}}(\xi_{t-1}^k)} \,. \tag{20}$$

9:   update

$$\hat{h}_t^{\phi_t,i} \leftarrow \sum_{j=1}^N \bar{w}_{t-1|t}^{\phi_t,i,j} \left( \hat{h}_{t-1}^{\phi_{t-1},j} + \ell_t^{\theta,\phi_t}(\xi_{t-1}^j, \xi_t^i) \right), \tag{21}$$

$$\hat{u}_t^{\phi_t,i} \leftarrow \sum_{i=1}^N \bar{w}_{t-1|t}^{\phi_t,i,j} \left\{ \hat{u}_{t-1}^{\phi_{t-1},j} + \nabla_\phi \log q_{t-1|t}^{\phi_t}(\xi_t^i, \xi_{t-1}^j) \left( \hat{h}_{t-1}^{\phi_{t-1},j} + \ell_t^{\theta,\phi_t}(\xi_{t-1}^j, \xi_t^i) \right) \right\}, \tag{22}$$

$$\hat{v}_t^{\theta_t,i} \leftarrow \sum_{i=1}^N \bar{w}_{t-1|t}^{\phi_t,i,j} \left\{ \hat{v}_{t-1}^{\theta_{t-1},j} + \nabla_\theta \ell_t^{\theta,\phi_t}(\xi_{t-1}^j, \xi_t^i) \right\} . \tag{23}$$

10:   Set

$$\widehat{\nabla}_\phi \mathcal{L}_t^{\theta_t,\phi_t} = \frac{1}{N} \sum_{i=1}^N \left\{ \nabla_\phi \log q_t^{\phi_t}(\xi_t^i) \hat{h}_t^{\phi_t,i} + \hat{u}_t^{\phi_t,i} \right\},$$

$$\widehat{\nabla}_\theta \mathcal{L}_t^{\theta_t,\phi_t} = \frac{1}{N} \sum_{i=1}^N \hat{v}_t^{\theta_t,i}.$$

11:   Update

$$\phi_{t+1} \leftarrow \phi_t + \gamma_{t+1}^\phi \left( \widehat{\nabla}_\phi \mathcal{L}_t^{\theta_t,\phi_t} - \widehat{\nabla}_\phi \mathcal{L}_{t-1}^{\theta_{t-1},\phi_{t-1}} \right), \tag{24}$$

$$\theta_{t+1} \leftarrow \theta_t + \gamma_{t+1}^\theta \left( \widehat{\nabla}_\theta \mathcal{L}_t^{\theta_t,\phi_t} - \widehat{\nabla}_\theta \mathcal{L}_{t-1}^{\theta_{t-1},\phi_{t-1}} \right) . \tag{25}$$

12: **end for**

---

## B. Proof of proposition 4.1

By definition,

$$\mathcal{L}_t^{\theta,\phi} = \mathbb{E}_{q_{0:t}^\phi} \left[ \log \frac{p_{0:t}^\theta(X_{0:t}, Y_{0:t})}{q_{0:t}^\phi(X_{0:t})} \right] = \mathbb{E}_{q_{0:t}^\phi} \left[ \log \frac{\chi(X_0) g_0^\theta(X_0, Y_0) \prod_{s=1}^t m_s^\theta(X_{s-1}, X_s) g_s^\theta(X_s, Y_s)}{q_t^\phi(X_t) \prod_{s=1}^t q_{s-1|s}^\phi(X_s, X_{s-1})} \right],$$

which, using (9), yields

$$\mathcal{L}_t^{\theta,\phi} = \mathbb{E}_{q_{0:t}^\phi} \left[ \sum_{s=0}^t \ell_s^{\theta,\phi}(X_{s-1}, X_s) - \log q_t^\phi(X_t) \right].$$

Therefore,
$$\mathcal{L}_t^{\theta,\phi} = \mathbb{E}_{q_t^\phi}\left[h_t(X_t)\right] - \mathbb{E}_{q_t^\phi}\left[\log q_t^\phi(X_t)\right], \tag{26}$$

where
$$h_t(x_t) = \int \left(\sum_{s=0}^t \ell_s^{\theta,\phi}(x_{s-1}, x_s)\right) \prod_{s=1}^t q_{s-1|s}^\phi(x_s, x_{s-1})\, \mathrm{d}x_{0:t-1} = \mathbb{E}_{q_{0:t-1|t}^\phi(x_t, \cdot)}\left[\ell_{0:t}^{\theta,\phi}(X_{0:t-1}, x_t)\right],$$

with $q_{0:t-1|t}^\phi(x_t, x_{0:t-1}) = \prod_{s=1}^t q_{s-1|s}^\phi(x_s, x_{s-1})$, and we introduced the notation $\ell_{0:t}^{\theta,\phi}(x_{0:t}) = \sum_{s=0}^t \ell_s^{\theta,\phi}(x_{s-1}, x_s)$. Then, note that

$$h_t(x_t) = \int \left(h_{t-1}(x_{t-1}) + \ell_t^{\theta,\phi}(x_{t-1}, x_t)\right) q_{t-1|t}^\phi(x_t, x_{t-1})\, \mathrm{d}x_{t-1}$$
$$= \mathbb{E}_{q_{t-1|t}^\phi(x_t, \cdot)}\left[h_{t-1}(X_{t-1}) + \ell_t^{\theta,\phi}(X_{t-1}, x_t)\right].$$

This establishes the recursive expression of the ELBO.

Now, we consider the gradient of the ELBO with respect to $\phi$. Write

$$\nabla_\phi \mathcal{L}_t^{\theta,\phi} = \nabla_\phi \mathbb{E}_{q_t^\phi}\left[h_t(X_t)\right] - \nabla_\phi \mathbb{E}_{q_t^\phi}\left[\log q_t^\phi(X_t)\right]$$
$$= \nabla_\phi \int \left(h_t(x_t) - \log q_t^\phi(x_t)\right) q_t^\phi(x_t)\, \mathrm{d}x_t$$
$$= \int \left(\nabla_\phi h_t(x_t) - \nabla_\phi \log q_t^\phi(x_t)\right) q_t^\phi(x_t)\, \mathrm{d}x_t + \int \left(h_t(x_t) - \log q_t^\phi(x_t)\right) \nabla_\phi q_t^\phi(x_t)\, \mathrm{d}x_t$$
$$= \mathbb{E}_{q_t^\phi}\left[\nabla_\phi h_t(X_t) + \left(h_t(X_t) - \log q_t^\phi(X_t)\right) \nabla_\phi \log q_t^\phi(X_t)\right],$$

where we used that $\mathbb{E}_{q_t^\phi}[\nabla \log q_t^\phi(X_t)] = 0$. Then, writing $u_t(x_t) = \nabla_\phi h_t(x_t)$,

$$u_t(x_t) = \nabla_\phi \mathbb{E}_{q_{0:t-1|t}^\phi(x_t, \cdot)}\left[\ell_{0:t}^{\theta,\phi}(X_{0:t-1}, x_t)\right]$$
$$= \mathbb{E}_{q_{0:t-1|t}^\phi(x_t, \cdot)}\left[\left(\nabla_\phi \log q_{0:t-1|t}^\phi \times \ell_{0:t}^{\theta,\phi}\right)(X_{0:t-1}, x_t)\right] + \mathbb{E}_{q_{0:t-1|t}^\phi(x_t, \cdot)}\left[\nabla_\phi \ell_{0:t}^{\theta,\phi}(X_{0:t-1}, x_t)\right].$$

Remembering that

$$\ell_{0:t}^{\theta,\phi} = \log\left(\chi(x_0) g_0^\theta(x_0, y_0)\right) + \sum_{s=1}^t \log\left(m_s^\theta(x_{s-1}, x_s) g_s^\theta(x_s, y_s)\right) - \sum_{s=1}^t \log q_{s-1|s}^\phi(x_s, x_{s-1}),$$

we obtain that $\nabla_\phi \ell_{0:t}^{\theta,\phi}(X_{0:t-1}, x_t) = -\nabla_\phi \log q_{0:(t-1)|t}^\phi(X_{0:t-1}, x_t)$, which has zero expectation under $q_{0:t-1|t}^\phi(x_t, \cdot)$. Thus,

$$u_t(x_t) = \mathbb{E}_{q_{0:t-1|t}^\phi(x_t, \cdot)}\left[\left(\nabla_\phi \log q_{0:(t-1)|t}^\phi \times \ell_{0:t}^{\theta,\phi}\right)(X_{0:t-1}, x_t)\right].$$

To establish the recursion for $u_t(x_t)$, write

$$u_t(x_t) = \mathbb{E}_{q_{0:t-1|t}^\phi(x_t, \cdot)}\left[\left(\nabla_\phi \log q_{0:(t-2)|t-1}^\phi(X_{0:t-1}) + \nabla_\phi \log q_{t-1|t}^\phi(X_{t-1}, x_t)\right)\right.$$
$$\left. \times \left(\ell_{0:t-1}^{\theta,\phi}(X_{0:t-1}) + \ell_t^{\theta,\phi}(X_{t-1}, x_t)\right)\right] \tag{27}$$
$$= \mathbb{E}_{q_{t-1|t}^\phi(x_t, \cdot)}\left[u_{t-1}(X_{t-1})\right] \tag{28}$$
$$+ \mathbb{E}_{q_{t-1|t}^\phi(x_t, \cdot)}\left[\nabla_\phi \log q_{t-1|t}^\phi(X_{t-1}, x_t)\left(\mathbb{E}_{q_{0:(t-2)|t-1}^\phi}\left[\ell_{0:t-1}^{\theta,\phi}(X_{0:t-1})\right] + \ell_t^{\theta,\phi}(X_{t-1}, x_t)\right)\right]$$
$$+ \mathbb{E}_{q_{t-1|t}^\phi(x_t, \cdot)}\left[\ell_t^{\theta,\phi}(X_{t-1}, x_t) \times \mathbb{E}_{q_{0:(t-2)|t-1}^\phi}\left[\nabla_\phi \log q_{0:(t-2)|t-1}^\phi(X_{0:t-1})\right]\right],$$

which yields

$$u_t(x_t) = \mathbb{E}_{q^\phi_{t-1|t}} \left[ u_{t-1}(X_{t-1}) + \nabla_\phi \log q^\phi_{t-1|t}(X_{t-1}, x_t) \times \left( h_{t-1}(X_{t-1}) + \ell^{\theta,\phi}_t(X_{t-1}, x_t) \right) \right],$$

which was to be established.

Finally, let us consider the gradient w.r.t. $\theta$. Using (26), we have that $\nabla_\theta \mathcal{L}^{\theta,\phi}_t = \mathbb{E}_{q^\phi_t} [\nabla_\theta h_t(X_t)]$. Writing $v_t = \nabla_\theta h_t$, we obtain

$$
\begin{aligned}
v_t(x_t) &= \nabla_\theta \mathbb{E}_{q^\phi_{0:t-1|t}(x_t,\cdot)} \left[ \ell^{\theta,\phi}_{0:t}(X_{0:t-1}, x_t) \right] \\
&= \mathbb{E}_{q^\phi_{0:t-1|t}(x_t,\cdot)} \left[ \nabla_\theta \ell^{\theta,\phi}_{0:t-1}(X_{0:t-2}, X_{t-1}) + \nabla_\theta \ell^{\theta,\phi}_{t-1}(X_{t-1}, x_t) \right] \\
&= \mathbb{E}_{q^\phi_{0:t-1|t}(x_t,\cdot)} \left[ \nabla_\theta \ell^{\theta,\phi}_{0:t-1}(X_{0:t-2}, X_{t-1}) \right] + \mathbb{E}_{q^\phi_{t-1|t}(x_t,\cdot)} \left[ \nabla_\theta \ell^{\theta,\phi}_{t-1}(X_{t-1}, x_t) \right] \\
&= \mathbb{E}_{q^\phi_{t-1|t}(x_t,\cdot)} \left[ v_{t-1}(X_{t-1}) \right] + \mathbb{E}_{q^\phi_{t-1|t}(x_t,\cdot)} \left[ \nabla_\theta \ell^{\theta,\phi}_{t-1}(X_{t-1}, x_t) \right],
\end{aligned}
$$

which concludes the proof.

## C. Proof of Theorem 4.5

### C.1. Geometric ergodicity of the extended chain

First, some notation. Let $(\mathsf{E}, \mathcal{E})$ be an arbitrary state space. Then a kernel $K$ on $\mathsf{E} \times \mathcal{E}$ induces two endomorphisms, the first acting on the space $\mathsf{F}_b(\mathcal{E})$ of bounded measurable functions on $(\mathsf{E}, \mathcal{E})$ according to

$$\mathsf{F}_b(\mathcal{E}) \ni f \mapsto Kf(\cdot) := \int f(x) \, K(\cdot, \mathrm{d}x) \in \mathsf{F}_b(\mathcal{E})$$

and the second acting on the space $\mathsf{M}(\mathcal{E})$ of measures on $(\mathsf{E}, \mathcal{E})$ (we let $\mathsf{M}_1(\mathcal{E}) \subset \mathsf{M}(\mathcal{E})$ denote the subspace of probability measures) according to

$$\mathsf{M}(\mathcal{E}) \ni \mu \mapsto \mu K(x, \cdot) := \int \mu(\mathrm{d}x) \, K(x, \cdot) \in \mathsf{M}(\mathcal{E}).$$

In addition, the product of two kernels $K$ and $L$ on $(\mathsf{E}, \mathcal{E})$ is defined as the kernel

$$KL : \mathsf{E} \times \mathcal{E} \ni (x, A) \mapsto \int K(x, \mathrm{d}x') \, L(x', A)$$

on $(\mathsf{E}, \mathcal{E})$. Using this notation, we may define, for any $t \in \mathbb{N}_{>0}$, the power $K^t$ of a kernel $K$ by multiplying $K$ by itself $t-1$ times.

In the following we denote, for every $t \in \mathbb{N}$, by

$$Q^\phi_{t|t+1} f(x_{t+1}) := \int f(x_t) q^\phi_{t|t+1}(x_t, x_{t+1}) \, \mathrm{d}x_t, \quad (x_{t+1}, f) \times \mathsf{F}_b(\mathcal{X}),$$

the Markov kernel induced by the transition density $q^\phi_{t|t+1}$.

In Section 4 it is assumed that the data generating process $(X_t, Y_t)_{t \in \mathbb{N}}$ is an SSM, and we denote by $R$ its Markov transition kernel of this process. Since the auxiliary states $(a_t)_{t \in \mathbb{N}}$ are generated deterministically from the observations via the mapping $\mathcal{A}^\phi$, also the augmented process $(X_t, Y_t, a_t)_{t \in \mathbb{N}}$ is Markov with transition kernel

$$S^\phi f(x_t, y_t, a_t) := \int f(x_{t+1}, y_{t+1}, \mathcal{A}^\phi(a_t, y_{t+1})) \, R(x_t, y_t, \mathrm{d}(x_{t+1}, y_{t+1})),$$

$$(x_t, y_t, a_t, f) \in \mathsf{X} \times \mathsf{Y} \times \mathsf{A} \times \mathsf{F}_b(\mathcal{X} \otimes \mathcal{Y} \otimes \mathcal{A}).$$

For clarity, we now briefly recall the principal assumptions of this work.

**Assumption C.1** (Uniform ergodicity of $(X_t, Y_t, a_t)_{t\in\mathbb{N}}$)**.** There exist $\pi \in \mathsf{M}_1(\mathcal{X} \otimes \mathcal{Y} \otimes \mathcal{A})$ and $\alpha \in (0,1)$ such that for every $t \in \mathbb{N}_{>0}$, $\phi \in \Phi$, and $(x, y, a) \in \mathsf{X} \times \mathsf{Y} \times \mathsf{A}$,

$$\|(S^\phi)^t(x, y, a) - \pi\|_{\mathsf{TV}} \le \alpha^t.$$

Assumption C.1 is discussed in Section C.2 below.

**Assumption C.2.** There exist constants $0 < \varepsilon^- < \varepsilon^+ < \infty$ such that for every $\phi \in \Phi$, $t \in \mathbb{N}_{>0}$, and $(x, x') \in \mathsf{X}^2$,

$$\varepsilon^- \le \psi_t^\phi(x, x') \le \varepsilon^+.$$

In addition, we let $\varepsilon := \varepsilon^- / \varepsilon^+$.

**Assumption C.3.** There exists $c \in \mathbb{R}_{>0}$ such that for every $\phi \in \Phi$, $a_t \in \mathsf{A}$, and $(x_{s+1}, y_{s+1}) \in \mathsf{X}^2 \times \mathsf{Y}$,

(i) $\displaystyle\int |\nabla_\phi \log q_{s|s+1}^\phi(x_{s+1}, x_s)|^2 \, Q_{s|s+1}^\phi(x_{s+1}, \mathrm{d}x_s) \le c^2,$

(ii) $\displaystyle\int |\ell_s^{\theta,\phi}(x_s, x_{s+1})|^2 \, Q_{s|s+1}^\phi(x_{s+1}, \mathrm{d}x_s) \le c^2,$

(iii) $\displaystyle\int |\nabla_\theta \ell_s^{\theta,\phi}(x_s, x_{s+1})| \, Q_{s|s+1}^\phi(x_{s+1}, \mathrm{d}x_s) \le c.$

From now on we let $\mathsf{Z} := \mathsf{X} \times \mathsf{Y} \times \mathsf{A} \times \mathsf{F}_\mathsf{b}(\mathcal{X})^3$ denote the state space of the extended chain $(Z_t)_{t\in\mathbb{N}}$ and let $\mathcal{Z} := \mathcal{X} \otimes \mathcal{Y} \otimes \mathcal{A} \otimes \mathcal{F}_\mathsf{b}(\mathcal{X})^{\otimes 3}$ be the associated $\sigma$-field.

**Definition C.4.** Let $\mathcal{L}(\mathcal{Z})$ be the set of $f \in \mathsf{F}(\mathcal{Z})$ for which there exists $\varphi \in \mathsf{F}_\mathsf{b}(\mathcal{X} \otimes \mathcal{Y} \otimes \mathcal{A})$ such that for every $(x, y, a, h, h', u, u', v, v') \in \mathsf{X} \times \mathsf{Y} \times \mathsf{A} \times \mathsf{F}_\mathsf{b}(\mathcal{X})^6$,

(i) $|f(x, y, a, h, u, v)| \le \varphi(x, y, a),$

(ii) $|f(x, y, a, h, u, v) - f(x, y, a, h', u', v')| \le \varphi(x, y, a) \, (\mathrm{osc}(h - h') + \mathrm{osc}(u - u') + \mathrm{osc}(v - v')).$

The following is ma slightly more precise statement of Theorem 4.5, our main result.

**Theorem C.5.** *Assume C.1, C.3, and C.2. Then there exist $\rho \in (0,1)$ and a function $\zeta : \mathsf{F}_\mathsf{b}(\mathcal{X})^6 \to \mathbb{R}_{>0}$ such that for every $(\theta, \phi) \in \Theta \times \Phi$, $t \in \mathbb{N}$, $f \in \mathcal{L}(\mathcal{Z})$, $z = (x_0, y_0, a_0, h_0, u_0, v_0) \in \mathsf{Z}$, and $z' = (x'_0, y'_0, a'_0, h'_0, u'_0, v'_0) \in \mathsf{Z}$,*

$$|(T^{\theta,\phi})^t f(z) - (T^{\theta,\phi})^t f(z')| \le \zeta(h_0, h'_0, u_0, u'_0, v_0, v'_0)\|\varphi\|_\infty \rho^t. \tag{29}$$

*Moreover, there exists a kernel $\Pi^{\theta,\phi}$ on $\mathsf{Z} \times \mathcal{L}(\mathcal{Z})$ such that for every $f \in \mathcal{L}(\mathcal{Z})$, $\Pi^{\theta,\phi} f$ is constant and for every $t \in \mathbb{N}$ and $z = (x_0, y_0, a_0, h_0, u_0, v_0) \in \mathsf{Z}$,*

$$|(T^{\theta,\phi})^t f(z) - \Pi^{\theta,\phi} f| \le \bar{\zeta}(h_0, u_0, v_0)\|\varphi\|_\infty \rho^t, \tag{30}$$

*where*

$$\bar{\zeta}(h_0, u_0, v_0) := \frac{1}{1-\rho} \int \zeta(h_0, h'_0, u_0, u'_0, v_0, v'_0) \, T^{\theta,\phi}(z, \mathrm{d}z'). \tag{31}$$

We preface the proof of Theorem C.5 with a couple of definitions and lemmas. First, we summarize the recursion for the ELBO using the function-valued mapping

$$\mathcal{H}^{\theta,\phi}(\cdot, a_t, y_{t+1}) : h_t \mapsto \int \left( h_t(x_t) + \ell_t^{\theta,\phi}(x_t, x_{t+1}) \right) Q_{t|t+1}^\phi(x_{t+1}, \mathrm{d}x_t),$$

where the kernel $Q_{t|t+1}^\phi$ and the term $\ell_t^{\theta,\phi}$ depend implicitly on $a_t$ and $y_{t+1}$, respectively, implying that $h_{t+1}(x_{t+1}) = \mathcal{H}^{\theta,\phi}(h_t, a_t, y_{t+1})(x_{t+1})$. Based on the latter, we also define, for every $t \in \mathbb{N}_{>0}$ and vector $y_{1:t} \in \mathsf{Y}^t$, the composite versions

$$\mathcal{H}^{\theta,\phi}(\cdot, a_{0:t-1}, y_{1:t}) : h_0 \mapsto \begin{cases} \mathcal{H}^{\theta,\phi}(h_0, a_0, y_1), & \text{for } t = 1, \\ \mathcal{H}^{\theta,\phi}(\mathcal{H}_{t-1}^{\theta,\phi}(h_0, a_{0:t-2}, y_{1:t-1}), a_{t-1}, y_t), & \text{for } t \ge 2. \end{cases} \tag{32}$$

Using the similar notation

$$\mathcal{V}^{\theta,\phi}(\cdot, v_t, a_t) : v_t \mapsto \int \left( v_t(x_t) + \nabla_\theta \ell_t^{\theta,\phi}(x_t, x_{t+1}) \right) Q_{t|t+1}^\phi(x_{t+1}, \mathrm{d}x_t),$$

for the recursion of the ELBO gradient with respect to $\theta$, the composite mappings $(\mathcal{V}_t^{\theta,\phi})_{t \in \mathbb{N}_{>0}}$ are defined similarly.

Finally, letting

$$\mathcal{U}^{\theta,\phi}(\cdot, a_t, y_{t+1}) : (h_t, u_t) \mapsto \int \left( u_t(x_t) + \nabla_\phi \log q_{t|t+1}^\phi(x_t, x_{t+1})\{h_t(x_t) + \ell_t^{\theta,\phi}(x_t, x_{t+1})\} \right) Q_{t|t+1}^\phi(x_{t+1}, \mathrm{d}x_t)$$

summarize the recursion for the ELBO gradient with respect to $\phi$, so that $u(x_{t+1}) = \mathcal{U}^{\theta,\phi}(h_t, u_t, a_t, y_{t+1})(x_{t+1})$, we also define the compositions

$$\mathcal{U}_t^{\theta,\phi}(\cdot, a_{0:t-1}, y_{1:t}) : (h_0, u_0) \mapsto \begin{cases} \mathcal{U}^{\theta,\phi}(h_0, u_0, a_0, y_1), & \text{for } t = 1, \\ \mathcal{U}^{\theta,\phi}(\mathcal{H}_{t-1}^{\theta,\phi}(h_0, a_{0:t-2}, y_{1:t-1}), \mathcal{U}_{t-1}^{\theta,\phi}(h_0, u_0, a_{0:t-2}, y_{1:t-1}), a_{t-1}, y_t), & \text{for } t \geq 2. \end{cases}$$
(33)

Using these definitions, the transition kernel $T^{\theta,\phi}$ of the extended chain $(Z_t)_{t \in \mathbb{N}}$ can be expressed as

$$T^{\theta,\phi} f(z_t) = \int f(x_{t+1}, y_{t+1}, \mathcal{A}^{\theta,\phi}(a_t, y_{t+1}), \mathcal{H}^{\theta,\phi}(h_t, a_t, y_{t+1}), \mathcal{U}^{\theta,\phi}(h_t, u_t, a_t, y_{t+1}), \mathcal{V}^{\theta,\phi}(v_t, a_t, y_{t+1}))$$
$$\times R((x_t, y_t), \mathrm{d}(x_{t+1}, y_{t+1})), \quad (z_t, f) \in \mathsf{Z} \times \mathsf{F_b}(\mathcal{Z}), \quad (34)$$

where $z_t = (x_t, y_t, a_t, h_t, u_t, v_t)$.

The following lemma establishes the geometric contraction of the function-valued mappings defined above.

**Lemma C.6.** *Assume C.2. Then for every* $t \in \mathbb{N}$, $y_{1:t} \in \mathsf{Y}^t$, $a_{0:t-1} \in \mathsf{A}^t$, *and* $(h_0, h_0', v_0, v_0') \in \mathsf{F_b}(\mathcal{X})^4$,

(i) $\mathrm{osc}\left( \mathcal{H}_t^{\theta,\phi}(h_0, a_{0:t-1}, y_{1:t}) - \mathcal{H}_t^{\theta,\phi}(h_0', a_{0:t-1}, y_{1:t}) \right) \leq (1 - \varepsilon)^t \, \mathrm{osc}(h_0 - h_0')$,

(ii) $\mathrm{osc}\left( \mathcal{V}_t^{\theta,\phi}(v_0, a_{0:t-1}, y_{1:t}) - \mathcal{V}_t^{\theta,\phi}(v_0', a_{0:t-1}, y_{1:t}) \right) \leq (1 - \varepsilon)^t \, \mathrm{osc}(v_0 - v_0')$,

*where* $\varepsilon \in (0, 1)$ *is given in Assumption C.2.*

(iii) *Assume additionally C.3 (i). Then for every* $\varrho \in (1 - \varepsilon, 1)$ *there exists* $d > 0$ *such that for every* $t \in \mathbb{N}$, $y_{1:t} \in \mathsf{Y}^t$, $a_{0:t-1} \in \mathsf{A}^t$, *and* $(h_0, h_0', u_0, u_0') \in \mathsf{F_b}(\mathcal{X})^4$,

$$\mathrm{osc}\left( \mathcal{U}_t^{\theta,\phi}(h_0, u_0, a_{0:t-1}, y_{1:t}) - \mathcal{U}_t^{\theta,\phi}(h_0', u_0', a_{0:t-1}, y_{1:t}) \right) \leq (1 - \varepsilon)^t \, \mathrm{osc}(u_0 - u_0') + d\varrho^t \, \mathrm{osc}(h_0 - h_0').$$

The proof of Lemma C.6 is based on the following lemmas.

**Lemma C.7.** *for every* $t \in \mathbb{N}_{>0}$, $y_{1:t} \in \mathsf{Y}^t$, $a_{0:t-1} \in \mathsf{A}^t$, $(h_0, u_0) \in \mathsf{F_b}(\mathcal{X})^2$, *and* $x_t \in \mathsf{X}$,

$$\mathcal{U}_t^{\theta,\phi}(h_0, u_0, a_{0:t-1}, y_{1:t})(x_t)$$
$$= \int \cdots \int \left( u_0(x_0) + \sum_{s=0}^{t-1} \nabla_\phi \log q_{s|s+1}^\phi(x_{s+1}, x_s)\{\mathcal{H}_s^{\theta,\phi}(h_0, a_{0:s-1}, y_{1:s}) + \ell_s^{\theta,\phi}(x_s, x_{s+1})\} \right) \prod_{s=0}^{t-1} Q_{s|s+1}^\phi(x_{s+1}, \mathrm{d}x_s)$$

*with the convention* $\mathcal{H}_0^{\theta,\phi}(h_0, a_{0:-1}, y_{1:0}) := h_0$.

*Proof of Lemma C.7.* We proceed by induction and assume that the claim holds true for $t \in \mathbb{N}_{>0}$. By definition (33),

$$\mathcal{U}_{t+1}^{\theta,\phi}(h_0, u_0, a_{0:t}, y_{1:t+1})(x_{t+1})$$
$$= \mathcal{U}^{\theta,\phi}(\mathcal{H}_t^{\theta,\phi}(h_0, a_{0:t-1}, y_{1:t}), \mathcal{U}_t^{\theta,\phi}(u_0, a_{0:t-1}, y_{1:t}), a_t, y_{t+1})$$
$$= \int \mathcal{U}_t^{\theta,\phi}(u_0, a_{0:t-1}, y_{1:t})(x_t) + \nabla_\phi \log q_{t|t+1}^\phi(x_{t+1}, x_t)\{\mathcal{H}_t^{\theta,\phi}(h_0, a_{0:t-1}, y_{1:t})(x_t) + \ell_t^{\theta,\phi}(x_t, x_{t+1})\} Q_{t|t+1}^\phi(x_{t+1}, \mathrm{d}x_t).$$

Now, inserting the induction hypothesis into the right-hand side of the previous expression yields

$$\mathcal{U}_{t+1}^{\theta,\phi}(h_0, u_0, a_{0:t}, y_{1:t+1})(x_{t+1})$$
$$= \int \cdots \int \left( u_0(x_0) + \sum_{s=0}^{t} \nabla_\phi \log q_{s|s+1}^\phi(x_{s+1}, x_s) \{ \mathcal{H}_s^{\theta,\phi}(h_0, a_{0:s-1}, y_{1:s}) + \ell_s^{\theta,\phi}(x_s, x_{s+1}) \} \right) \prod_{s=1}^{t} Q_{s|s+1}^\phi(x_{s+1}, \mathrm{d}x_s),$$

which establishes the induction step.

Finally, we note that the base case $t = 1$ holds true, since by definition (33)

$$\mathcal{U}_1^{\theta,\phi}(h_0, u_0, a_0, y_1)(x_1) = \mathcal{U}^{\theta,\phi}(h_0, u_0, a_0, y_1)(x_1)$$
$$= \int \left( u_0(x_0) + \nabla_\phi \log q_{0|1}^\phi(x_1, x_0) \{ h_0(x_0) + \ell_0^{\theta,\phi}(x_0, x_{0+1}) \} \right) Q_{0|1}^\phi(x_1, \mathrm{d}x_0).$$

This completes the proof. $\qquad\square$

In the following, let $\beta(M)$ denote the Dobrushin coefficient of a Markov kernel $M$.

**Lemma C.8.** *Assume C.2. Then for every* $t \in \mathbb{N}$, $\beta(Q_{t|t+1}^\phi) \leq 1 - \varepsilon$.

*Proof of Lemma C.8.* Pick arbitrarily $(x_{t+1}, f) \in \mathsf{X} \times \mathsf{F}_\mathsf{b}(\mathcal{X})$ and write, using definition (6) and Assumption C.2,

$$Q_{t|t+1}^\phi f(x_{t+1}) = \frac{\int f(x_t) \psi_t^\phi(x_t, x_{t+1}) q_t^\phi(\mathrm{d}x_t)}{\int \psi_t^\phi(x_t', x_{t+1}) q_t^\phi(\mathrm{d}x_t')} \geq \frac{\varepsilon^-}{\varepsilon^+} q_t^\phi f = \varepsilon q_t^\phi f,$$

which means that $Q_{t|t+1}^\phi$ allows $\mathsf{X}$ as a 1-small set with respect to $(q_t^\phi, \varepsilon)$. From this it follows that $\beta(Q_{t|t+1}^\phi) \leq 1 - \varepsilon$. $\quad\square$

We are now ready to establish Lemma C.6.

*Proof of Lemma C.6.* To establish (i), let $(Q_{s-1|s}^\phi)_{s=1}^t$ denote the backward transition kernels associated with $(q_{s-1|s}^\phi)_{s=1}^t$. We may then write, for every $x_t \in \mathsf{X}$,

$$\mathcal{H}_t^{\theta,\phi}(h_0, y_{1:t})(x_t) - \mathcal{H}_t^{\theta,\phi}(h_0', y_{1:t})(x_t) = Q_{t-1|t}^\phi \cdots Q_{0|1}^\phi(h_0 - h_0')(x_t).$$

Now, recall that for every $s$ and $h \in \mathsf{F}_\mathsf{b}(\mathcal{X})$,

$$\mathrm{osc}(Q_{s-1|s}^\phi h) \leq \beta(Q_{s-1|s}^\phi) \, \mathrm{osc}(h), \tag{35}$$

where $\beta(Q_{s-1|s}^\phi)$ is the Dobrushin coefficient of $Q_{s-1|s}^\phi$, and iterating the bound (35) yields

$$\mathrm{osc}(Q_{t-1|t}^\phi \cdots Q_{0|1}^\phi(h_0 - h_0')) \leq \left( \prod_{s=1}^t \beta(Q_{s-1|s}^\phi) \right) \mathrm{osc}(h_0 - h_0'). \tag{36}$$

From this the claim (i) follows by Lemma C.8.

To prove (ii), note that

$$\mathcal{V}_t^{\theta,\phi}(v_0, a_{0:t-1}, y_{1:t})(x_t) - \mathcal{V}_t^{\theta,\phi}(v_0', a_{0:t-1}, y_{1:t})(x_t) = Q_{t-1|t}^\phi \cdots Q_{0|1}^\phi(v_0 - v_0')(x_t)$$
$$= \mathcal{H}_t^{\theta,\phi}(v_0, a_{0:t-1}, y_{1:t})(x_t) - \mathcal{H}_t^{\theta,\phi}(v_0', a_{0:t-1}, y_{1:t})(x_t).$$

Thus, (ii) follows immediately from (i).

Finally, to establish (iii), write, using Lemma C.7,

$$\mathcal{U}_t^{\theta,\phi}(h_0, u_0, a_{0:t-1}, y_{1:t})(x_t) - \mathcal{U}_t^{\theta,\phi}(h_0', u_0', a_{0:t-1}, y_{1:t})(x_t)$$

$$= \int \cdots \int \left( u_0(x_0) - u_0'(x_0) + \sum_{s=0}^{t-1} \nabla_\phi \log q_{s|s+1}^\phi(x_{s+1}, x_s) \{ \mathcal{H}_s^{\theta,\phi}(h_0, a_{0:s-1}, y_{1:s})(x_s) - \mathcal{H}_s^{\theta,\phi}(h_0', a_{0:s-1}, y_{1:s})(x_s) \} \right)$$

$$\times \prod_{s=0}^{t-1} Q_{s|s+1}^\phi(x_{s+1}, \mathrm{d}x_s)$$

$$= Q_{t-1|t}^\phi \cdots Q_{0|1}^\phi (u_0 - u_0')(x_t) + \sum_{s=0}^{t-1} Q_{t-1|t}^\phi \cdots Q_{s+1|s+2}^\phi \varphi_s(x_t), \tag{37}$$

where we have set

$$\varphi_s(x_{s+1}) := \int \nabla_\phi \log q_{s|s+1}^\phi(x_{s+1}, x_s) \{ \mathcal{H}_s^{\theta,\phi}(h_0, a_{0:s-1}, y_{1:s})(x_s) - \mathcal{H}_s^{\theta,\phi}(h_0', a_{0:s-1}, y_{1:s})(x_s) \} Q_{s|s+1}^\phi(x_{s+1}, \mathrm{d}x_s).$$

Now, note that since

$$\int \nabla_\phi \log q_{s|s+1}^\phi(x_{s+1}, x_s) Q_{s|s+1}^\phi(x_{s+1}, \mathrm{d}x_s) = 0,$$

it holds, for every $c \in \mathbb{R}$,

$$\|\varphi_s\|_\infty \leq \|\mathcal{H}_s^{\theta,\phi}(h_0, a_{0:s-1}, y_{1:s}) - \mathcal{H}_s^{\theta,\phi}(h_0', a_{0:s-1}, y_{1:s}) - c\|_\infty \int |\nabla_\phi \log q_{s|s+1}^\phi(x_{s+1}, x_s)| Q_{s|s+1}^\phi(x_{s+1}, \mathrm{d}x_s).$$

Thus, using Assumption C.3(i) and the fact that for all $f \in \mathsf{F}(\mathcal{X})$, $\mathrm{osc}(f) = 2 \inf_{c \in \mathbb{R}} \|f - c\|_\infty$, it holds, by (i), that

$$\|\varphi_s\|_\infty \leq \frac{1}{2} c \, \mathrm{osc} \left( \mathcal{H}_s^{\theta,\phi}(h_0, a_{0:s-1}, y_{1:s}) - \mathcal{H}_s^{\theta,\phi}(h_0', a_{0:s-1}, y_{1:s}) \right) \leq \frac{1}{2} c (1 - \varepsilon)^s \, \mathrm{osc}(h_0 - h_0').$$

As a consequence, by Lemma C.8,

$$\mathrm{osc}(Q_{t-1|t}^\phi \cdots Q_{s+1|s+2}^\phi \varphi_s) \leq \left( \prod_{\ell=s+1}^{t-1} \beta(Q_{\ell|\ell+1}^\phi) \right) \mathrm{osc}(\varphi_s)$$

$$\leq c(1 - \varepsilon)^{t-s-1}(1 - \varepsilon)^s \, \mathrm{osc}(h_0 - h_0')$$

$$= c(1 - \varepsilon)^{t-1} \, \mathrm{osc}(h_0 - h_0'). \tag{38}$$

Moreover, since

$$Q_{t-1|t}^\phi \cdots Q_{0|1}^\phi (u_0 - u_0')(x_t) = \mathcal{H}_t^{\theta,\phi}(u_0, a_{0:t-1}, y_{1:t})(x_t) - \mathcal{H}_t^{\theta,\phi}(u_0', a_{0:t-1}, y_{1:t})(x_t),$$

(i) implies that

$$\mathrm{osc}(Q_{t-1|t}^\phi \cdots Q_{0|1}^\phi (u_0 - u_0')) \leq (1 - \varepsilon)^t \, \mathrm{osc}(u_0 - u_0'). \tag{39}$$

Combining (37), (38), and (39) yields

$$\mathrm{osc} \left( \mathcal{U}_t^{\theta,\phi}(h_0, u_0, a_{0:t-1}, y_{1:t}) - \mathcal{U}_t^{\theta,\phi}(h_0', u_0', a_{0:t-1}, y_{1:t}) \right) \leq (1 - \varepsilon)^t \, \mathrm{osc}(u_0 - u_0') + ct(1 - \varepsilon)^{t-1} \, \mathrm{osc}(h_0 - h_0').$$

Finally, the claim (iii) follows by picking $\varrho \in (1 - \varepsilon, 1)$ and letting $d := ((1 - \varepsilon) \, \mathrm{e} \log\{\varrho/(1 - \varepsilon)\})^{-1}$. $\qquad\square$

We are now ready to establish Theorem C.5, following the same lines as Proposition C.12.

*Proof of Theorem C.5.* First, denote

$$f_{s,t}(x_t, y_{s:t}, a_{s-1:t}, h, u, v)$$

$$:= f(x_t, y_t, a_t, \mathcal{H}_{t-s+1}^{\theta,\phi}(h, a_{s-1:t-1}, y_{s:t}), \mathcal{U}_{t-s+1}^{\theta,\phi}(h, u, a_{s-1:t-1}, y_{s:t}), \mathcal{V}_{t-s+1}^{\theta,\phi}(v, a_{s-1:t-1}, y_{s:t})).$$

for $s \in [\![1, t]\!]$, where we have omitted the dependence on $\theta$ and $\phi$ for brevity. Note that with this notation, for $z = (x_0, y_0, a_0, h_0, u_0, v_0)$,

$$(T^{\theta,\phi})^t f(z) = \int \cdots \int f_{1,t}(x_t, y_{1:t}, a_{0:t}, h_0, u_0, v_0) \prod_{s=0}^{t-1} K^{\theta,\phi}((x_s, y_s, a_s), \mathrm{d}(x_{s+1}, y_{s+1}, a_{s+1})).$$

Now, picking $(\tilde{h}_0, \tilde{u}_0, \tilde{v}_0) \in \mathsf{F}_\mathsf{b}(\mathcal{X})^3$ arbitrarily and using the decomposition

$$f_{1,t}(x_t, y_{1:t}, a_{0:t}, h_0, u_0, v_0) = f_{1,t}(x_t, y_{1:t}, a_{0:t}, h_0, u_0, v_0) - f_{1,t}(x_t, y_{1:t}, a_{0:t}, \tilde{h}_0, \tilde{u}_0, \tilde{v}_0)$$

$$+ \sum_{s=1}^{t-1} \Big( f_{s,t}(x_t, y_{s:t}, a_{s-1:t}, \tilde{h}_0, \tilde{u}_0, \tilde{v}_0) - f_{s+1,t}(x_t, y_{s+1:t}, a_{s:t}, \tilde{h}_0, \tilde{u}_0, \tilde{v}_0) \Big) + f_{t,t}(x_t, y_t, a_{t-1:t}, \tilde{h}_0, \tilde{u}_0, \tilde{v}_0),$$

which is adopted from (Tadić & Doucet, 2005), we may write,

$(T^{\theta,\phi})^t f(z) - (T^{\theta,\phi})^t f(z')$

$$= \int \cdots \int \Big( f_{1,t}(x_t, y_{1:t}, a_{0:t}, h_0, u_0, v_0) - f_{1,t}(x_t, y_{1:t}, a_{0:t}, \tilde{h}_0, \tilde{u}_0, \tilde{v}_0) \Big) \prod_{s=0}^{t-1} S^\phi((x_s, y_s, a_s), \mathrm{d}(x_{s+1}, y_{s+1}, a_{s+1}))$$

$$- \int \cdots \int \Big( f_{1,t}(x'_t, y'_{1:t}, a'_{0:t}, h'_0, u'_0, v'_0) - f_{1,t}(x'_t, y'_{1:t}, a'_{0:t}, \tilde{h}_0, \tilde{u}_0, \tilde{v}_0) \Big) \prod_{s=0}^{t-1} S^\phi((x'_s, y'_s, a'_s), \mathrm{d}(x'_{s+1}, y'_{s+1}, a'_{s+1}))$$

$$+ \sum_{s=1}^{t} \int \cdots \int \Big( f_{s,t}(x_t, y_{s:t}, a_{s-1:t}, \tilde{h}_0, \tilde{u}_0, \tilde{v}_0) - f_{s+1,t}(x_t, y_{s+1:t}, a_{s:t}, \tilde{h}_0, \tilde{u}_0, \tilde{v}_0) \Big)$$

$$\times \Delta S_s^\phi((x_0, y_0, a_0), \mathrm{d}(x_s, y_s, a_s)) \prod_{\ell=s}^{t-1} S^\phi((x_\ell, y_\ell, a_\ell), \mathrm{d}(x_{\ell+1}, y_{\ell+1}, a_{\ell+1}))$$

$$- \sum_{s=1}^{t} \int \cdots \int \Big( f_{s,t}(x_t, y_{s:t}, a_{s-1:t}, \tilde{h}_0, \tilde{u}_0, \tilde{v}_0) - f_{s+1,t}(x_t, y_{s+1:t}, a_{s:t}, \tilde{h}_0, \tilde{u}_0, \tilde{v}_0) \Big)$$

$$\times \Delta S_s^\phi((x'_0, y'_0, a'_0), \mathrm{d}(x_s, y_s, a_s)) \prod_{\ell=s}^{t-1} S^\phi((x_\ell, y_\ell, a_\ell), \mathrm{d}(x_{\ell+1}, y_{\ell+1}, a_{\ell+1}))$$

$$+ \int f_{t,t}(x_t, y_t, a_{t-1:t}, \tilde{h}_0, \tilde{u}_0, \tilde{v}_0) \Delta S_t^\phi((x_0, y_0, a_0), \mathrm{d}(x_t, y_t, a_t))$$

$$- \int f_{t,t}(x_t, y_t, a_{t-1:t}, \tilde{h}_0, \tilde{u}_0, \tilde{v}_0) \Delta S_t^\phi((x'_0, y'_0, a'_0), \mathrm{d}(x_t, y_t, a_t)), \tag{40}$$

where we have defined, for $s \in [\![1, t]\!]$, the signed kernels

$$\Delta S_s^\phi f(x, y, a) := (S^\phi)^s f(x, y, a) - \pi f, \quad (f, x, y, a) \in \mathsf{F}_\mathsf{b}(\mathcal{X} \otimes \mathcal{Y} \otimes \mathcal{A}) \times \mathsf{X} \times \mathsf{Y} \times \mathsf{A}.$$

Note that by Definition C.11, for every $s \in [\![1, t]\!]$,

$$|f_{s,t}(x_t, y_{s:t}, a_{s-1:t}, \tilde{h}_0, \tilde{u}_0, \tilde{v}_0) - f_{s+1,t}(x_t, y_{s+1:t}, a_{s:t}, \tilde{h}_0, \tilde{u}_0, \tilde{v}_0)|$$

$$\leq \varphi(x_t, y_t, a_t) \Big( \mathrm{osc}(\mathcal{H}_{t-s+1}^{\theta,\phi}(\tilde{h}_0, a_{s-1:t-1}, y_{s:t}) - \mathcal{H}_{t-s}^{\theta,\phi}(\tilde{h}_0, a_{s:t-1}, y_{s+1:t}))$$

$$+ \mathrm{osc} \Big( \mathcal{U}_{t-s+1}^{\theta,\phi}(\tilde{h}_0, \tilde{u}_0, a_{s-1:t-1}, y_{s:t}) - \mathcal{U}_{t-s}^{\theta,\phi}(\tilde{h}_0, \tilde{u}_0, a_{s:t-1}, y_{s+1:t}) \Big)$$

$$+ \mathrm{osc} \Big( \mathcal{V}_{t-s+1}^{\theta,\phi}(\tilde{v}_0, a_{s-1:t-1}, y_{s:t}) - \mathcal{V}_{t-s}^{\theta,\phi}(\tilde{v}_0, a_{s:t-1}, y_{s+1:t}) \Big) \Big). \tag{41}$$

Using definition (32) and Lemma C.6(i), we conclude that

$$\mathrm{osc} \Big( \mathcal{H}_{t-s+1}^{\theta,\phi}(\tilde{h}_0, a_{s-1:t-1}, y_{s:t}) - \mathcal{H}_{t-s}^{\theta,\phi}(\tilde{h}_0, a_{s:t-1}, y_{s+1:t}) \Big)$$

$$= \mathrm{osc} \Big( \mathcal{H}_{t-s}^{\theta,\phi}(\mathcal{H}^{\theta,\phi}(\tilde{h}_0, y_s), a_{s:t-1}, y_{s+1:t}) - \mathcal{H}_{t-s}^{\theta,\phi}(\tilde{h}_0, a_{s:t-1}, y_{s+1:t}) \Big)$$

$$\leq (1 - \varepsilon)^{t-s} \, \mathrm{osc} \Big( \mathcal{H}^{\theta,\phi}(\tilde{h}_0, y_s) - \tilde{h}_0 \Big).$$

Since

$$\|\mathcal{H}^{\theta,\phi}(\tilde{h}_0, y_s) - \tilde{h}_0\|_\infty = \left\|\int (\tilde{h}_0(x) + \ell_{s-1}^{\theta,\phi}(x,\cdot))\, Q_{s-1|s}^\phi(\cdot, \mathrm{d}x) - \tilde{h}_0(\cdot)\right\|_\infty$$

$$\leq 2\|\tilde{h}_0\|_\infty + c,$$

where the constant $c \in \mathbb{R}_{>0}$ is provided by Assumption C.3(ii), it holds that

$$\mathrm{osc}\left(\mathcal{H}_{t-s+1}^{\theta,\phi}(\tilde{h}_0, a_{s-1:t-1}, y_{s:t}) - \mathcal{H}_{t-s}^{\theta,\phi}(\tilde{h}_0, a_{s:t-1}, y_{s+1:t})\right) \leq 2(2\|\tilde{h}_0\|_\infty + c)(1-\varepsilon)^{t-s}. \tag{42}$$

By the same arguments it is shown that

$$\mathrm{osc}\left(\mathcal{V}_{t-s+1}^{\theta,\phi}(\tilde{v}_0, a_{s-1:t-1}, y_{s:t}) - \mathcal{V}_{t-s}^{\theta,\phi}(\tilde{v}_0, a_{s:t-1}, y_{s+1:t})\right) \leq 2(2\|\tilde{v}_0\|_\infty + c)(1-\varepsilon)^{t-s}. \tag{43}$$

In addition, similarly, using Lemma C.6(iii),

$$\mathrm{osc}\left(\mathcal{U}_{t-s+1}^{\theta,\phi}(\tilde{h}_0, \tilde{u}_0, a_{s-1:t-1}, y_{s:t}) - \mathcal{U}_{t-s}^{\theta,\phi}(\tilde{h}_0, \tilde{u}_0, a_{s:t-1}, y_{s+1:t})\right)$$

$$= \mathrm{osc}\left(\mathcal{U}_{t-s}^{\theta,\phi}(\mathcal{H}^{\theta,\phi}(\tilde{h}_0, y_s), \mathcal{U}^{\theta,\phi}(\tilde{h}_0, \tilde{u}_0, y_s), a_{s:t-1}, y_{s+1:t}) - \mathcal{U}_{t-s}^{\theta,\phi}(\tilde{h}_0, \tilde{u}_0, a_{s:t-1}, y_{s+1:t})\right)$$

$$\leq (1-\varepsilon)^{t-s}\, \mathrm{osc}\left(\mathcal{H}^{\theta,\phi}(\tilde{h}_0, y_s) - \tilde{h}_0\right) + d\varrho^{t-s}\, \mathrm{osc}\left(\mathcal{U}^{\theta,\phi}(\tilde{h}_0, \tilde{u}_0, y_s) - \tilde{u}_0\right),$$

and since by Assumption C.3(i–ii) and the Cauchy–Schwarz inequality,

$$\left\|\mathcal{U}^{\theta,\phi}(\tilde{h}_0, \tilde{u}_0, y_s) - \tilde{u}_0\right\|_\infty = \left\|\int (\tilde{u}_0(x) + \nabla_\phi \log q_{s-1|s}^\phi(\cdot, x)\{\tilde{h}_0(x) + \ell_{s-1}^{\theta,\phi}(x,\cdot)\})\, Q_{s-1|s}^\phi(\cdot, \mathrm{d}x) - \tilde{h}_0(\cdot)\right\|_\infty$$

$$\leq 2\|\tilde{u}_0\|_\infty + c\|\tilde{h}_0\|_\infty + c^2,$$

we may conclude that

$$\mathrm{osc}\left(\mathcal{U}_{t-s+1}^{\theta,\phi}(\tilde{h}_0, \tilde{u}_0, a_{s-1:t-1}, y_{s:t}) - \mathcal{U}_{t-s}^{\theta,\phi}(\tilde{h}_0, \tilde{u}_0, a_{s:t-1}, y_{s+1:t})\right)$$

$$\leq 2(2\|\tilde{h}_0\|_\infty + c)(1-\varepsilon)^{t-s} + 2d(2\|\tilde{u}_0\|_\infty + c\|\tilde{h}_0\|_\infty + c^2)\varrho^{t-s}. \tag{44}$$

Combining (41), (42), (43), and (44) yields the bound

$$|f_{s,t}(x_t, y_{s:t}, a_{s-1:t}, \tilde{h}_0, \tilde{u}_0, \tilde{v}_0) - f_{s+1,t}(x_t, y_{s+1:t}, a_{s:t}, \tilde{h}_0, \tilde{u}_0, \tilde{v}_0)|$$

$$\leq 2\varphi(x_t, y_t, a_t)\left(3(2\|\tilde{h}_0\|_\infty + c)(1-\varepsilon)^{t-s} + d(2\|\tilde{u}_0\|_\infty + c\|\tilde{h}_0\|_\infty + c^2)\varrho^{t-s}\right)$$

$$\leq 2\varphi(x_t, y_t, a_t)\left((6+cd)\|\tilde{h}_0\|_\infty + 2d\|\tilde{u}_0\|_\infty + 3c + c^2 d\right)\varrho^{t-2}. \tag{45}$$

Similarly,

$$|f_{1,t}(x_t, y_{1:t}, a_{0:t}, h_0, u_0, v_0) - f_{1,t}(x_t, y_{1:t}, a_{0:t}, \tilde{h}_0, \tilde{u}_0, \tilde{v}_0)| \leq \varphi(x_t, y_t, a_t)\zeta''(h_0, u_0, v_0)\varrho^t, \tag{46}$$

where we have defined the mapping

$$\zeta''(h_0, u_0, v_0) := 2\left((d+1)(\|h_0\| + \|\tilde{h}_0\|_\infty) + \|u_0\| + \|\tilde{u}_0\|_\infty + \|v_0\| + \|\tilde{v}_0\|_\infty\right).$$

Now, letting

$$\zeta'(h_0, h_0', u_0, u_0', v_0, v_0') := 2\left((6+cd)\|\tilde{h}_0\|_\infty + 2d\|\tilde{u}_0\|_\infty + 3c + c^2 d\right) \vee \zeta''(h_0, u_0, v_0) \vee \zeta''(h_0', u_0', v_0').$$

Applying the bounds (45) and (46) to the decomposition (40) yields

$$|(T^{\theta,\phi})^t f(z) - (T^{\theta,\phi})^t f(z')| \leq \zeta'(h_0, h'_0, u_0, u'_0, v_0, v'_0)\varrho^t \left((S^\phi)^t \varphi(x_0, y_0, a_0) + (S^\phi)^t \varphi(x'_0, y'_0, a'_0)\right)$$

$$+ \zeta'(h_0, h'_0, u_0, u'_0, v_0, v'_0) \sum_{s=1}^t \varrho^{t-s} \left(\Delta S_s^\phi (S^\phi)^{t-s}\varphi(x_0, y_0, a_0) + \Delta S_s^\phi (S^\phi)^{t-s}\varphi(x'_0, y'_0, a'_0)\right)$$

$$+ \Delta S_t^\phi \varphi(x_0, y_0, a_0) + \Delta S_t^\phi \varphi(x'_0, y'_0, a'_0). \quad (47)$$

Now, by applying Assumption C.1 to the right-hand side of (57) we obtain

$$|(T^{\theta,\phi})^t f(z) - (T^{\theta,\phi})^t f(z')| \leq \|\varphi\|_\infty \left(\zeta'(h_0, h'_0, u_0, u'_0, v_0, v'_0)\alpha^t + \zeta'(h_0, h'_0, u_0, u'_0, v_0, v'_0)t(\alpha \vee \varrho)^t + \varrho^t\right),$$

from which (50) follows by picking $\rho \in (\alpha \vee \varrho, 1)$ and

$$\zeta(h_0, h'_0, u_0, u'_0, v_0, v'_0) := 2(\zeta'(h_0, h'_0, u_0, u'_0, v_0, v'_0)+1)(\alpha \vee \varrho)/\rho + 2\zeta'(h_0, h'_0, u_0, u'_0, v_0, v'_0) \left(e \,|\log\{(\alpha \vee \varrho)/\rho\}|\right)^{-1}.$$

To prove the second claim, first note that by (50),

$$|(T^{\theta,\phi})^{t+1} f(z) - (T^{\theta,\phi})^t f(z)| \leq \int |(T^{\theta,\phi})^t f(z') - (T^{\theta,\phi})^t f(z)| \, T^{\theta,\phi}(z, dz')$$

$$\leq \|\varphi\|_\infty \rho^t \int \zeta(h_0, h'_0, u_0, u'_0, v_0, v'_0) \, T^{\theta,\phi}(z, dz'). \quad (48)$$

Now, define the kernel

$$\Pi^{\theta,\phi} f(z) := f(z) + \sum_{t=0}^\infty \left((T^{\theta,\phi})^{t+1} f(z) - (T^{\theta,\phi})^t f(z)\right)$$

on $\mathsf{Z} \times \mathcal{L}(\mathcal{Z})$. With this definition, note that by (58),

$$|(T^{\theta,\phi})^t f(z) - \Pi^{\theta,\phi} f(z)| \leq \sum_{s=t}^\infty |(T^{\theta,\phi})^{s+1} f(z) - (T^{\theta,\phi})^s f(z)|$$

$$\leq \bar{\zeta}(h_0, u_0, v_0)\|\varphi\|_\infty \rho^t,$$

where $\bar{\zeta}(h_0, u_0, v_0)$ is defined in (31), which establishes (51). Finally, it remains to prove that the function $\Pi^{\theta,\phi} f$ is constant. For this purpose, pick arbitrarily $(z, z') \in \mathsf{Z}^2$; then, however, by (50) and (51),

$$|\Pi^{\theta,\phi} f(z) - \Pi^{\theta,\phi} f(z')| \leq \inf_{t \in \mathbb{N}} \left(|(T^{\theta,\phi})^{t+1} f(z) - (T^{\theta,\phi})^t f(z)| + |(T^{\theta,\phi})^t f(z) - \Pi^{\theta,\phi} f(z)|\right.$$

$$\left. + |(T^{\theta,\phi})^t f(z') - \Pi^{\theta,\phi} f(z')|\right) = 0,$$

from which the claim follows. $\qquad\square$

### C.2. Discussion on Assumption 4.2

Assumption 4.2 assumes uniform ergodicity of $(X_t, Y_t, a_t)_{t \in \mathbb{N}}$. In this section, we present two sufficient conditions under which this assumption applies (as proven by Propositon C.12), if not for all $f \in \mathsf{F_b}(\mathcal{X} \otimes \mathcal{Y} \otimes \mathcal{A})$, then at least for all $f$ in a certain Lipschitz subclass $\tilde{\mathcal{L}}(\mathcal{X} \otimes \mathcal{Y} \otimes \mathcal{A})$ of $\mathsf{F}(\mathcal{X} \otimes \mathcal{Y} \otimes \mathcal{A})$ to be specified.

A first assumption is that the Markov chain induced by the model is itself uniformly ergodic, formally:

**Assumption C.9** (Uniform ergodicity of $(X_t, Y_t)_{t \in \mathbb{N}}$)**.** There exist $\beta \in (0, 1)$ and $\sigma \in \mathsf{M}_1(\mathcal{X} \otimes \mathcal{Y})$ such that for every $(x, y) \in \mathsf{X} \times \mathsf{Y}$ and $t \in \mathbb{N}_{>0}$,

$$\|R^t((x, y), \cdot) - \sigma\|_{\mathsf{TV}} \leq \beta^t,$$

where $R((x, y), \cdot)$ is the Markov kernel of the state space model.

This assumption is classical in the recursive maximum likelihood setting.

In addition to this, we suppose, in our online variational setting, a contraction property for the the mapping $\mathcal{A}^\phi$ introduced in Section 3. Let's recall that, for every $t \in \mathbb{N}_{>0}$ and vectors $y_{1:t} \in \mathsf{Y}^t$ and $a_{0:t-1} \in \mathsf{A}^t$,

$$\mathcal{A}_t^\phi(\cdot, y_{1:t}) := \begin{cases} \mathcal{A}^\phi(\cdot, y_1), & \text{for } t = 1, \\ \mathcal{A}^\phi(\mathcal{A}_{t-1}^\phi(\cdot, y_{1:t-1}), y_t), & \text{for } t \geq 2. \end{cases} \tag{49}$$

We consider the following assumption.

**Assumption C.10.** There exist $c > 0$ and $\kappa \in (0, 1)$ such that for every $\phi \in \Phi$, $t \in \mathbb{N}$, $y_{1:t} \in \mathsf{Y}^t$, and $(a, \tilde{a}) \in \mathsf{A}^2$,

(i) $\|\mathcal{A}_t^\phi(a, y_{1:t}) - \mathcal{A}_t^\phi(\tilde{a}, y_{1:t})\|_2 \leq c\|a - \tilde{a}\|_2 \kappa^t$,

(ii) $\|\mathcal{A}^\phi(a, y_t) - a\|_2 \leq c$.

As discussed below (Remark C.13) this assumption is relatively mild and satisfied by on-the-shelf recurrent neural networks.

Now we can show that Assumptions C.9-C.10 implies Assumption 4.2. We first introduces some definitions.

**Definition C.11.** Let $\tilde{\mathcal{L}}(\mathcal{X} \otimes \mathcal{Y} \otimes \mathcal{A})$ be the set of $f \in \mathsf{F}(\mathcal{X} \otimes \mathcal{Y} \otimes \mathcal{A})$ for which there exists $\tilde{\varphi} \in \mathsf{F}_\mathsf{b}(\mathcal{X} \otimes \mathcal{Y})$ such that for every $(x, y, a, a') \in \mathsf{X} \times \mathsf{Y} \times \mathsf{A}^2$,

(i) $|f(x, y, a)| \leq \tilde{\varphi}(x, y)$,

(ii) $|f(x, y, a) - f(x, y, a')| \leq \tilde{\varphi}(x, y)\|a - a'\|_2$.

Then the following Proposition links C.9 and C.10 to Assumption 4.2.

**Proposition C.12.** *Assume C.9 and C.10. Then there exists a function $\delta : \mathsf{A}^2 \to \mathbb{R}_{>0}$ and $\alpha \in (0, 1)$ such that for every $\phi \in \Phi$, $t \in \mathbb{N}$, $f \in \tilde{\mathcal{L}}(\mathcal{X} \otimes \mathcal{Y} \otimes \mathcal{A})$, and $((x, y, a), (x', y', a')) \in (\mathsf{X} \times \mathsf{Y} \times \mathsf{A})^2$,*

$$|(S^\phi)^t f(x, y, a) - (S^\phi)^t f(x', y', a')| \leq \delta(a, a')\|\tilde{\varphi}\|_\infty \alpha^t. \tag{50}$$

*Moreover, there exists a kernel $\pi^\phi$ on $(\mathsf{X} \times \mathsf{Y} \times \mathsf{A}) \times \tilde{\mathcal{L}}(\mathcal{X} \otimes \mathcal{Y} \otimes \mathcal{A})$ such that for every $f \in \mathcal{L}(\mathcal{X} \otimes \mathcal{Y} \otimes \mathcal{A})$, $\pi^\phi f$ is constant and for every $t \in \mathbb{N}$ and $(x, y, a) \in \mathsf{X} \times \mathsf{Y} \times \mathsf{A}$,*

$$|(S^\phi)^t f(x, y, a) - \pi^\phi f| \leq \bar{\delta}(a)\|\tilde{\varphi}\|\alpha^t, \tag{51}$$

*where*

$$\bar{\delta}(a) := \frac{1}{1 - \alpha} \int \delta(a, a') \, S^\phi((x, y, a), \mathrm{d}(x', y', a')). \tag{52}$$

Before proving the results, we briefly comment Assumption C.10.

*Remark* C.13 (About Assumption C.10). To illustrate that Assumption C.10(i) is a mild, assumption, let $\mathcal{A}^\phi$ correspond to the following vanilla recurrent neural network architecture, where $a_0 \in \mathbb{R}^d$ is the initial hidden state. For $t \in \mathbb{N}_{>0}$, given a sequence $y_{1:t}$ of observations, the hidden state $a_t$ is given by

$$a_t = \mathcal{A}^\phi(a_{t-1}, y_t) = \sigma(W_\phi a_{t-1} + U_\phi y_t + b_\phi),$$

where

- $W_\phi \in \mathbb{R}^{d \times d}$ and $U_\phi \in \mathbb{R}^{d \times d_y}$ are weight matrices and $b_\phi \in \mathbb{R}^d$ is a bias vector.

- $\sigma : \mathbb{R}^d \to \mathbb{R}^d$ is a Lipschitz mapping with Lipschitz constant $L_\sigma$. Typical examples include the ReLU and hyperbolic tangent (tanh) functions, both of which satisfy $L_\sigma = 1$.

Consider the spectral norm of $W_\phi$ given by

$$L_{W_\phi} = \|W_\phi\|_2 = \sup_{\|x\|_2 = 1} \|W_\phi x\|_2 \,,$$

*i.e.*, $L_{W_\phi}$ is the largest singular value of $W_\phi$. Now, let us consider two sequences $(a_t)_{t \in \mathbb{N}}$ and $(\tilde{a}_t)_{t \in \mathbb{N}}$ of hidden states starting with $a_0$ and $\tilde{a}_0$, respectively. For every fixed observation sequence $y_{1:t}$, it holds, by definition (49), that

$$\|\mathcal{A}_t^\phi(a_0, y_{1:t}) - \mathcal{A}_t^\phi(\tilde{a}_0, y_{1:t})\|_2 \leq L_\sigma \left\| W \left( \mathcal{A}_{t-1}^\phi(a_0, y_{1:t-1}) - \mathcal{A}_{t-1}^\phi(\tilde{a}_0, y_{1:t-1}) \right) \right\|_2$$
$$\leq L_\sigma^t \|W\|_2^t \|a_0 - \tilde{a}_0\|_2 \,.$$

Typically, we may assume here that $L_\sigma = 1$ (again, this is the case for the ReLU and tanh functions). Assumption C.10(i) then holds if $\|W_\phi\|_2 < 1$. This contraint can be enforced during training via spectral norm regularization (Yoshida & Miyato, 2017). Alternatively, one may enforce it *a priori* by fixing the spectral norm to be at most $\rho_{\max} < 1$ and, at each gradient step $k$, projecting the unconstrained weight matrix $\tilde{W}_\phi^{(k)}$ onto the corresponding spectral-norm ball according to

$$W_\phi^{(k)} = \rho_{\max} \frac{\tilde{W}_\phi^{(k)}}{\|\tilde{W}_\phi^{(k)}\|_2}.$$

Such projections are already standard in echo state networks (Sun et al., 2020).

*Proof of Proposition C.12.* We proceed as in the proof of Theorem C.5. To prove the first claim, we introduce the short-hand notation

$$f_{s,t}(x_t, y_{s:t}, a) := f(x_t, y_t, \mathcal{A}_{t-s+1}^\phi(a, y_{s:t})) \tag{53}$$

for $s \in [\![1, t]\!]$, where we have omitted the dependence on $\phi$ for brevity. Note that with this notation,

$$(S^\phi)^t f(z) = \int \cdots \int f_{1,t}(x_t, y_{1:t}, a) \, R((x, y), \mathrm{d}(x_1, y_1)) \prod_{s=1}^{t-1} R((x_s, y_s), \mathrm{d}(x_{s+1}, y_{s+1})).$$

Now, picking $\tilde{a} \in \mathsf{A}$ arbitrarily and using the decomposition

$$f_{1,t}(x_t, y_{1:t}, a) = f_{1,t}(x_t, y_{1:t}, a) - f_{1,t}(x_t, y_{1:t}, \tilde{a}) + \sum_{s=1}^{t} (f_{s,t}(x_t, y_{s:t}, \tilde{a}) - f_{s+1,t}(x_t, y_{s+1:t}, \tilde{a})) + f_{t,t}(x_t, y_t, \tilde{a}),$$

we may write,

$$(S^\phi)^t f(x, y, a) - (S^\phi)^t f(x', y', a')$$
$$= \int \cdots \int (f_{1,t}(x_t, y_{1:t}, a) - f_{1,t}(x_t, y_{1:t}, \tilde{a})) \, R((x, y), \mathrm{d}(x_1, y_1)) \prod_{s=1}^{t-1} R((x_s, y_s), \mathrm{d}(x_{s+1}, y_{s+1}))$$
$$- \int \cdots \int (f_{1,t}(x_t, y_{1:t}, a') - f_{1,t}(x_t, y_{1:t}, \tilde{a})) \, R((x', y'), \mathrm{d}(x_1, y_1)) \prod_{s=1}^{t-1} R((x_s, y_s), \mathrm{d}(x_{s+1}, y_{s+1}))$$
$$+ \sum_{s=1}^{t} \int \cdots \int (f_{s,t}(x_t, y_{s:t}, \tilde{a}) - f_{s+1,t}(x_t, y_{s+1:t}, \tilde{a})) \, \Delta R_s((x, y), \mathrm{d}(x_s, y_s)) \prod_{\ell=s}^{t-1} R((x_\ell, y_\ell), \mathrm{d}(x_{\ell+1}, y_{\ell+1}))$$
$$- \sum_{s=1}^{t} \int \cdots \int (f_{s,t}(x_t, y_{s:t}, \tilde{a}) - f_{s+1,t}(x_t, y_{s+1:t}, \tilde{a})) \, \Delta R_s((x', y'), \mathrm{d}(x_s, y_s)) \prod_{\ell=s}^{t-1} R((x_\ell, y_\ell), \mathrm{d}(x_{\ell+1}, y_{\ell+1}))$$
$$+ \int f_{t,t}(x_t, y_t, \tilde{a}) \, \Delta R_t((x, y), \mathrm{d}(x_t, y_t)) - \int f_{t,t}(x_t, y_t, \tilde{a}) \, \Delta R_t((x', y'), \mathrm{d}(x_t, y_t)), \tag{54}$$

where we have defined, for $s \in [\![1, t]\!]$, the signed kernels

$$\Delta R_s f(x, y) := R^s f(x, y) - \sigma f, \quad (f, x, y) \in \mathsf{F}_\mathsf{b}(\mathcal{X} \otimes \mathcal{Y}) \times \mathsf{X} \times \mathsf{Y}.$$

Note that by Definition C.11 and Assumption C.10, for every $s \in [\![1, t]\!]$,

$$
\begin{aligned}
|f_{s,t}(x_t, y_{s:t}, \tilde{a}) - f_{s+1,t}(x_t, y_{s+1:t}, \tilde{a})| &\leq \tilde{\varphi}(x_t, y_t) \|\mathcal{A}^\phi_{t-s+1}(\tilde{a}, y_{s:t}) - \mathcal{A}^\phi_{t-s}(\tilde{a}, y_{s+1:t})\|_2 \\
&= \tilde{\varphi}(x_t, y_t) \|\mathcal{A}^\phi_{t-s}(\mathcal{A}^\phi(\tilde{a}, y_s), y_{s+1:t}) - \mathcal{A}^\phi_{t-s}(\tilde{a}, y_{s+1:t})\|_2 \\
&\leq \tilde{\varphi}(x_t, y_t) c \|\mathcal{A}^\phi(\tilde{a}, y_s) - \tilde{a}\|_2 \kappa^{t-s} \\
&\leq \tilde{\varphi}(x_t, y_t) c^2 \kappa^{t-s}.
\end{aligned}
\tag{55}
$$

Similarly,

$$
\begin{aligned}
|f_{1,t}(x_t, y_{1:t}, a) - f_{1,t}(x_t, y_{1:t}, \tilde{a})| &\leq \tilde{\varphi}(x_t, y_t) c \|a - \tilde{a}\|_2 \\
&\leq \tilde{\varphi}(x_t, y_t) c (\|a\|_2 + \|\tilde{a}\|) \kappa^t.
\end{aligned}
\tag{56}
$$

Now, let

$$
\delta'(a, a') := c^2 \vee c(\|a\|_2 + \|\tilde{a}\|) \vee c(\|a'\|_2 + \|\tilde{a}\|);
$$

then, applying the bounds (55) and (56) to the decomposition (54) yields

$$
\begin{aligned}
&|(S^\phi)^t f(x, y, a) - (S^\phi)^t f(x', y', a')| \\
&\leq \delta'(a, a') \kappa^t \left( R^t \tilde{\varphi}(x, y) + R^t \tilde{\varphi}(x', y') \right) + \delta'(a, a') \sum_{s=1}^t \kappa^{t-s} \left( \Delta R_s R^{t-s} \tilde{\varphi}(x, y) + \Delta R_s R^{t-s} \tilde{\varphi}(x', y') \right) \\
&\hspace{8cm} + \Delta R_t \tilde{\varphi}(x, y) + \Delta R_t \tilde{\varphi}(x', y').
\end{aligned}
\tag{57}
$$

Now, by applying Assumption C.9 to the right-hand side of (57) we obtain

$$
|(S^\phi)^t f(x, y, a) - (S^\phi)^t f(x', y', a')| \leq 2\|\tilde{\varphi}\|_\infty \left( \delta'(a, a') \kappa^t + \delta'(a, a') t(\kappa \vee \beta)^t + \beta^t \right),
$$

from which (50) follows by picking $\alpha \in ((\kappa \vee \beta), 1)$ and letting

$$
\delta(a, a') := 2(\delta'(a, a') + 1)(\kappa \vee \beta)/\alpha + 2\delta'(a, a')(\mathrm{e} \, |\log\{(\kappa \vee \beta)/\alpha\}|)^{-1}.
$$

To prove the second claim, first note that by (50),

$$
\begin{aligned}
&|(S^\phi)^{t+1} f(x, y, a) - (S^\phi)^t f(x, y, a)| \\
&\leq \int |(S^\phi)^t f(x', y', a') - (S^\phi)^t f(x, y, a)| \, S^\phi((x, y, a), \mathrm{d}(x', y', a')) \\
&\leq \|\tilde{\varphi}\|_\infty \alpha^t \int \delta(a, a') \, S^\phi((x, y, a), \mathrm{d}(x', y', a')).
\end{aligned}
\tag{58}
$$

Now, define the kernel

$$
\pi^\phi f(x, y, a) := f(x, y, a) + \sum_{t=0}^\infty \left( (S^\phi)^{t+1} f(x, y, a) - (S^\phi)^t f(x, y, a) \right)
$$

on $\mathsf{X} \times \mathsf{Y} \times \mathsf{A} \times \mathcal{L}(\mathcal{X} \otimes \mathcal{Y} \otimes \mathcal{A})$. With this definition, note that by (58),

$$
\begin{aligned}
|(S^\phi)^t f(x, y, a) - \pi^\phi f(x, y, a)| &\leq \sum_{s=t}^\infty |(S^\phi)^{s+1} f(x, y, a) - (S^\phi)^s f(x, y, a)| \\
&\leq \delta(a) \|\tilde{\varphi}\|_\infty \alpha^t,
\end{aligned}
$$

where $\delta(a)$ is provided by (52), which establishes (51).

Finally, it remains to prove that the function $\pi^\phi f$ is constant. For this purpose, pick arbitrarily $((x, y, a), (x'y', a')) \in (\mathsf{X} \times \mathsf{Y} \times \mathsf{A})^2$; then, however, by (50) and (51),

$$
\begin{aligned}
|\pi^\phi f(x, y, a) - \pi^\phi f(x', y', a')| \leq \inf_{t \in \mathbb{N}} \big( &|(S^\phi)^{t+1} f(x, y, a) - (S^\phi)^t f(x, y, a)| \\
&+ |(S^\phi)^t f(x, y, a) - \pi^\phi f(x, y, a)| \\
&\hspace{2cm} + |(S^\phi)^t f(x', y', a') - \pi^\phi f(x', y', a')| \big) = 0,
\end{aligned}
$$

from which the claim follows. $\qquad\square$

## D. Full algorithm using backward sampling and control variate

In this section, we detail Algorithm 3, which presents one iteration of the online gradient ascent algorithm in the amortized scheme, incorporating both backward sampling and control variates for improved efficiency and variance reduction.

**Backward sampling.** Computing the backward weights of (20) has the disadvantage of $O(N^2)$ complexity due to the computation of the normalizing constant, which can be prohibitive when $N$ is large (typically for high-dimensional state spaces). One solution, suggested by Olsson & Westerborn (2017) in the context of SMC smoothing, is to use a backward sampling approach. More precisely, at time step $t$, given $\xi_t^i$, one samples independently $M$ indexes $\{j_k\}_{k=1}^M$ from the categorical distribution over $\{1, \ldots, N\}$ with weights $\{\bar{w}_{t-1|t}^{\phi,i,j}\}_{j=1}^N$, and replace (21) by $\sum_{k=1}^M (\hat{h}_{t-1}^{\phi_{t-1},j_k} + \ell_t^{\theta,\phi_t}(\xi_{t-1}^{j_k}, \xi_t^i))/M$. Olsson & Westerborn (2017) show that even with $M$ much smaller than $N$ (typically, $M = 2$), which provides a considerable improvement in complexity, this alternative estimator has only slightly higher variance than the original estimator. Here, noting that $\bar{w}_{t-1|t}^{\phi,i,j} \propto_j \psi_t^\phi(\xi_{t-1}^j, \xi_t^i)$, backward sampling can, in the case of bounded potential functions, be performed using an accept-reject procedure without having to calculate the normalizing constant of the weights. We refer the reader to (Olsson & Westerborn, 2017; Gloaguen et al., 2022; Dau & Chopin, 2023) for details and alternative backward sampling approaches.

**Variance reduction of the gradient estimator.** Proposition 4.1 involves computing *score-function expectations* in the form $\mathbb{E}_{q^\phi}\left[\nabla_\phi \log q^\phi(X) \cdot f(X)\right]$ for some p.d.f. $q^\phi$. As shown in Mohamed et al. (2020), direct Monte Carlo estimation of the score function leads to high variance and should normally not be used without a suitable variance reduction technique. The most straightforward approach is to design a *control variate*. Using the fact that $\mathbb{E}_{q^\phi}[\nabla_\phi \log q^\phi(X)] = 0$, the target expectation can be rewritten as $\mathbb{E}_{q^\phi}[\nabla_\phi \log q^\phi(X)\{f(X) - \mathbb{E}_{q^\phi}[f(X)]\}]$, which can be estimated with lower variance using a Monte Carlo estimate of $\mathbb{E}_{q^\phi}[f(X)]$. In our case, the latter is formed as a by-product of Algorithm 2, and therefore our methodology comes with built-in variance reduction without the need to recompute additional quantities. This accelerated version of Algorithm 2 is described in detail in Algorithm 3 (see Appendix D), which also includes the backward sampling technique described above.

As an alternative to this variance reduction technique, it is natural to consider the reparametrization trick, as it often leads to Monte Carlo estimators with lower variance compared to those obtained using the score function. However, the implementation of the reparametrization trick in this context requires that $\nabla_\phi \mathcal{L}_t^{\theta,\phi}$ is expressed as an expectation with respect to a random variable $Z_{0:t}$ that does not depend on $\phi$. Moreover, the recursive expression of this expectation at time $t + 1$ must be derivable from its predecessor, which is non-trivial. For example, in the classical case where $q_{0:t}$ is the p.d.f. of a multivariate Gaussian random variable with mean $\mu$ and variance $\Sigma$, and the expectation is taken w.r.t. $Z_{0:t} \sim \mathcal{N}(0, I_{d_x \times (t+1)})$, such a recursion is not feasible as the ELBO is no longer an additive functional when $X_{0:t}$ is replaced by $\mu + \Sigma^{\frac{1}{2}} Z_{0:t}$.

**On the use of Adam.** Our theoretical analysis (Section 4) is framed within the Robbins–Monro stochastic approximation setting. In practice we use Adam (Kingma & Ba, 2015) for all experiments, as it yields more stable performance; we did not observe SNR issues that would favor the decaying-step-size Robbins–Monro scheme.

## E. Supplementary details for the numerical experiments in Section 6

### E.1. Appendix for section 6.1 the linear Gaussian SSM.

**Variational Family.** In the linear Gaussian setting, the variational marginals $q_t^\phi$ are parameterized as Gaussian distributions defined by their natural parameters $\eta_t$. To ensure the backward kernel $q_{t-1|t}^\phi$ remains in the same Gaussian family, the potential is explicitly defined as

$$\psi_t^\phi(x_{t-1}, x_t) = \exp\left(\langle \tilde{\eta}_t^\phi(x_t), T(x_{t-1}) \rangle\right). \tag{59}$$

This formulation allows the backward density to be derived analytically by simply summing natural parameters, avoiding the need for normalizing constants. The parameters for the backward kernel are updated according to:

$$\eta_{t-1|t}^\phi = \eta_{t-1}^\phi + \tilde{\eta}_t^\phi. \tag{60}$$

Unlike the general case requiring neural networks, the recursions for these parameters in the linear Gaussian case are analytical, effectively mirroring the smoothing distribution updates of a standard linear Gaussian SSM.

*Table 2.* Steady-state MAE on the 2D linear-Gaussian SSM (last 20,000 steps).

| Method | Transition $F$ | Emission $G$ |
|---|---|---|
| RMCVI (Ours) | **0.0258** | **0.0408** |
| OSIWAE | 0.0416 | 0.0668 |
| OVSMC | 0.0416 | 0.0650 |

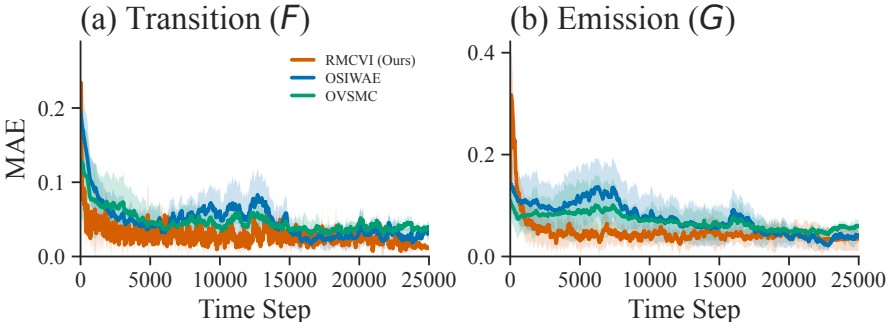

*Figure 6.* Parameter MAE on the 2D linear-Gaussian SSM for RMCVI, OSIWAE, and OVSMC: (a) transition $F$, (b) emission $G$.

**Parameters for the linear Gaussian SSM.** For the streaming experiment presented in the main text ($T = 50,000$), we learn both model and variational parameters from random initialization. We utilize learning rates of $10^{-3}$ for the variational parameters and $10^{-4}$ for the model parameters. To ensure a rigorous comparison, we replicate the generative settings of Campbell et al. (2021), using diagonal noise covariance matrices with fixed variances of $0.1$ for the transition and $0.25$ for the emission.

**Comparison with OVSMC and OSIWAE** We additionally benchmark RMCVI against two particle-based online variational methods on a 2D linear-Gaussian SSM: OVSMC (Mastrototaro et al., 2025) and OSIWAE (Mastrototaro & Olsson, 2024). Both rely on sequential resampling, which can cause path degeneracy and limits GPU parallelization. OSIWAE's IWAE objective further degrades the gradient signal-to-noise ratio. RMCVI instead uses i.i.d. samples and explicitly models the backward kernels (Eq. **??**), giving degeneracy-free trajectories with high-SNR gradients. As shown in Figure 6 and Table 2, RMCVI converges faster and reaches lower steady-state error for both parameters.

**Oracle ELBO** As an oracle baseline, we can compute the closed-form ELBO and its associated gradient via the reparameterization trick. Figure 7 displays the evolution of the ELBO in the case of the linear SSM and the offline setting, *i.e.*, when observations are processed through multiple epochs. In this specific experiment, we do not learn the model parameter. For our recursive method, we choose $\Delta = 2$ to truncate the backpropagation, as we observe that $\Delta < 2$ prevents our method from converging altogether, while $\Delta > 2$ only improves convergence speed by a small margin. The experiment is run using 10 different parameters for the generative model, $d_x = d_y = 10$, $T = 500$, and $N = 2$ for the two methods involving Monte Carlo sampling. It shows the convergence of our score-based solution to the correct optimum given by the analytical computations. This is particularly appealing and notably demonstrates that our online gradient-estimation method may perform well using few samples. In practice, we observe that the variance reduction introduced in Section D is crucial in reaching such performance.

### E.2. Appendix for section 6.2 the chaotic RNN

The 1-step smoothing and filtering errors in table 1 are given as

$$\kappa_T^{(1)} = \frac{1}{T-1} \sum_{t=1}^{T-1} \left( \frac{1}{d_x} \sum_{k=1}^{d_x} \left( \widehat{\mathbb{E}}_{q_{t-1:t}^\phi} \left[ X_{t-1}^{(k)} \right] - x_{t-1}^{*(k)} \right)^2 \right)^{1/2} \tag{61}$$

$$\kappa_T^{(2)} = \frac{1}{T} \sum_{t=1}^{T} \left( \frac{1}{d_x} \sum_{k=1}^{d_x} \left( \widehat{\mathbb{E}}_{q_t^\phi} \left[ X_t^{(k)} \right] - x_t^{*(k)} \right)^2 \right)^{1/2}, \tag{62}$$

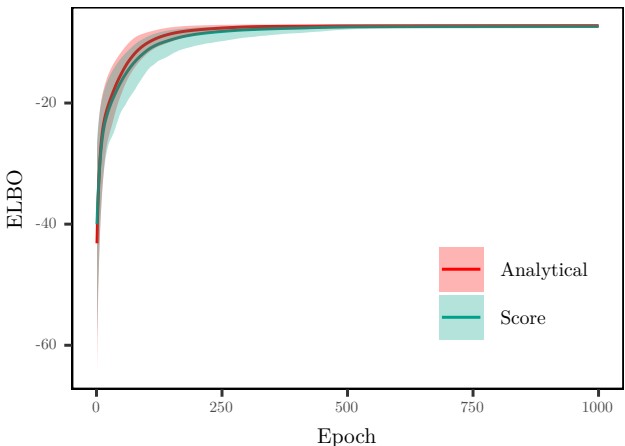

*Figure 7.* Evolution of $\mathcal{L}_T^{\theta,\phi}/T$ computed with three different methods and with three different types of gradients estimates. Full lines: means of the 10 replicates. Shaded lines: standard deviations of the 10 replicates.

| Gradients | $\Delta_{T,p}^\phi$ ($\times 10^{-2}$) | Avg. time |
|---|---|---|
| Score-based | $13.5 \pm 0.7$ (**12.2**) | 173 ms |
| Backward sampling | $11.9 \pm 0.4$ (**11.4**) | 17 ms |

*Table 3.* RMSE of the predicted marginal means $\mathbb{E}_{q_{0:T}^\phi}[X_t]$ w.r.t. the true states $x_t^*$ and average time per gradient step.

where $\widehat{\mathbb{E}}_q$ denotes the standard Monte Carlo estimate of an expectation w.r.t. $q$. Table 4 reports smoothing and filtering RMSE with respect to the true states at the end of training.

**Parameters for the chaotic RNN**  We choose the same generative hyperparameters as Campbell et al. (2021) with $\Delta = 0.001$, $\tau = 0.025$, $\gamma = 2.5$, 2 degrees of freedom and a scale of 0.1 for the Student-$t$ distribution, and define $Q = \text{diag}(0.01)$. For the joint learning setting, training is performed using the Adam (Kingma & Ba, 2015) optimizer with learning rates of $10^{-3}$ for the variational parameters and $10^{-4}$ for the model parameters. The variational backward kernels follow the parameterization defined in the main text, where the potential functions are parameterized by MLPs with 100 hidden units and tanh activations.

**Implementation settings for the comparison with Campbell et al. (2021)**  In the original paper, the variational distribution is designed in a non-amortized scheme, meaning that the variational parameters are not shared through time. Specifically, we have that $\phi = \{\phi^0, \dots, \phi^t, \dots, \}$, and each $\phi^t$ contains the parameter $\eta_t = (\mu_t, \Sigma_t)$ of the distribution $q_t^\phi \sim \mathcal{N}(\mu_t, \Sigma_t)$ and the parameter $\tilde{\eta}_t$ of the function $\psi_t^{\phi_t}$. For this latter function, we match the number of parameters of Campbell et al. (2021) by defining $\psi_t^\phi(x, y) = \exp\left(\tilde{\eta}_t(y) \cdot T(x)\right)$ with $\tilde{\eta}_t(y) = (\tilde{\eta}_{t,1}(y), \tilde{\eta}_{t,2})$ where $y \mapsto \tilde{\eta}_{t,1}(y)$ is a multi-layer perceptron with 100 neurons from $\mathbb{R}^{d_x}$ to $\mathbb{R}^{d_x}$, and $\tilde{\eta}_{t,2}$ is a negative definite matrix. We follow the optimization schedules of Campbell et al. (2021) with $K = 500$ gradient steps at each time-step.

**Modifications induced by the non-amortized scheme**  In the non-amortized scheme, $\phi = \{\phi^0, \dots, \phi^t\}$ is a set of distinct parameters, each parameter corresponding to a specific time step. In the notations of the article, the estimate $\phi_{t-1}$ of $\phi$ after having processed observations the $y_{0:t-1}$ depends on $\{\phi^0, \dots, \phi^{t-1}\}$. Therefore, the gradient of the ELBO will be w.r.t. $\phi^t$ only. This affects the expression of the statistic $u_t$, and one can see in the expansion of Eqn. (27) that the term (28) will now, when the gradient is taken w.r.t. $\phi^t$, be zero. This means that this term no longer has to be propagated. Indeed, as we set $q_{t-1|t}^{\phi_t}(x_t, x_{t-1}) \propto q_{t-1}^{\phi_{t-1}}(x_{t-1})\psi_t^{\phi_t}(x_{t-1}, x_t)$, the gradient of the ELBO w.r.t. $\phi^t$ will be

$$\nabla_{\phi^t}\mathcal{L}_t^{\theta,\phi_t} = \mathbb{E}_{q_t^{\phi_t}}\left[\mathbb{E}_{q_{t-1|t}^{\phi_t}}\left[\nabla \log q_t^\phi(X_t) \times \ell_t^{\theta,\phi_t}(X_{t-1}, X_t)\right.\right.$$
$$\left.\left. + \nabla_{\phi^t} \log q_{t-1|t}^{\phi_t}(X_{t-1}, x_t) \times \left(h_{t-1}(X_{t-1}) + \ell_t^{\theta,\phi_t}(X_{t-1}, X_t)\right)\right]\right].$$

| Sequence | Smoothing RMSE | Filtering RMSE |
|----------|---------------|----------------|
| Training | 0.281 | 0.311 |
| Eval | 0.278 ($\pm$ 0.01) | 0.305 ($\pm$ 0.014) |

*Table 4.* Smoothing and filtering RMSE when $\phi$ is learned online together with $(\rho, \gamma)$ in the chaotic RNN. Results are shown for the training stream and for independent sequences from the same generative model.

This gradient will be estimated using Monte Carlo in the same way as in the algorithm. In (Campbell et al., 2021), the inner conditional expectation is estimated with the regression approach (that we briefly recall below) instead of our importance sampling approach.

**Functional regression approach of Campbell et al. (2021)** Here, we recall the alternate option used in Campbell et al. (2021) to propagate approximations of the backward expectations. Denoting $\mathcal{F} = \left\{ g : \mathbb{R}^p \to \mathbb{R}^{d_x}, \mathbb{E}_{q_t^\phi}[\|g(X_t)\|_2] < \infty \right\}$, $h_t(x)$ satisfies, by definition of conditional expectation,

$$h_t = \operatorname*{argmin}_{g \in \mathcal{F}} \mathbb{E}_{q_{t-1:t}^\phi(x_{t-1}, x_t)} \|g(X_t) - [h_{t-1}(X_{t-1}) + \ell_t^{\theta,\phi}(X_{t-1}, X_t)]\|_2,$$

which provides a regressive objective for learning an approximation of $h_t$. In practice, the authors restrict the minimization problem to a subset of $\mathcal{F}$, a parametric family of functions (typically, a neural network) parameterized by $\gamma$ belonging to $\Gamma \subset \mathbb{R}^{d_\gamma}$, and learn this by approximating the expectation using Monte Carlo sampling. More precisely, the authors propose to estimate $h_t$ by $H_{\hat{\gamma}_t}^\phi$, where

$$\hat{\gamma}_t = \operatorname*{argmin}_{\gamma \in \Gamma} \frac{1}{N} \sum_{k=1}^{N} \|H_\gamma^\phi(\xi_t^k) - [H_{\hat{\gamma}_{t-1}}^\phi(\xi_{t-1}^k) + \tilde{h}_t(\xi_{t-1}^k, \xi_t^k)]\|_2, \tag{63}$$

where $\left\{ (\xi_{t-1}^i, \xi_t^i) \right\}_{i=1}^{N}$ is an i.i.d. sample from the variational joint distribution of $(X_{t-1}, X_t)$, which has density $q_{t-1:t}^\phi = q_t^\phi q_{t-1|t}^\phi$. Upon convergence, $H_{\hat{\gamma}_t}^\phi$ is then used in the successive recursions (similar to (21)).

### E.3. Appendix for section 6.3

**Air-quality data** The UCI Air Quality dataset contains hourly readings from 5 metal oxide chemical sensors located in a significantly polluted area. Our observation vector $y_t$ comprises Carbon Monoxide (CO), Non-Methane Hydrocarbons (NMHC), Nitrogen Oxides (NO$_x$, NO$_2$), Benzene (C$_6$H$_6$), Temperature (T), Relative Humidity (RH), and Absolute Humidity (AH). A key challenge of this dataset is the non-uniform data censorship caused by sensor failures. As shown in Figure 8, these dropouts occur at different intervals for different sensors. In the raw data, these are marked by a sentinel value ($-200$), which we treat as missing values (NaNs) during preprocessing.

**Parameters for the Air Quality Data** The transition function $f_\theta$ and emission function $g_\theta$ are both parameterized by MLPs with 2 hidden layers of 32 units each and $\tanh$ activation functions. The noise covariances $Q_\theta$ and $R_\theta$ are learned diagonal matrices. Regarding the variational approximation, the backward potentials are parameterized by MLPs with 32 hidden units and $\tanh$ activations. We use $N = 20$ importance samples for the Monte Carlo estimates. Training is performed using the Adam optimizer with learning rates of $10^{-3}$ for the variational parameters and $10^{-4}$ for the model parameters.

### E.4. Effective sample size diagnostics

A natural concern with importance-sampling-based estimators is weight degeneracy, usually monitored via the effective sample size $\text{ESS}_t = (\sum_i \bar{w}_{t-1|t}^{\phi,i})^2 / \sum_i (\bar{w}_{t-1|t}^{\phi,i})^2$. To check that this is not an issue in our setting, we tracked $\text{ESS}_t$ throughout training on the two experiments that bracket our sample-size regime: the linear-Gaussian SSM (Section 6.1) with $N = 100$, and the Air Quality data (Section 6.3) with only $N = 20$ samples.

Figure 9 shows $\text{ESS}_t$ as a percentage of $N$ over the full sequence. In both cases the ESS stays comfortably above the usual 10% collapse threshold for essentially the whole run. On the linear-Gaussian model, the minimum ESS observed is 16 out of 100, and the median is around 50%. The Air Quality experiment is the more demanding setting, small $N$, missing

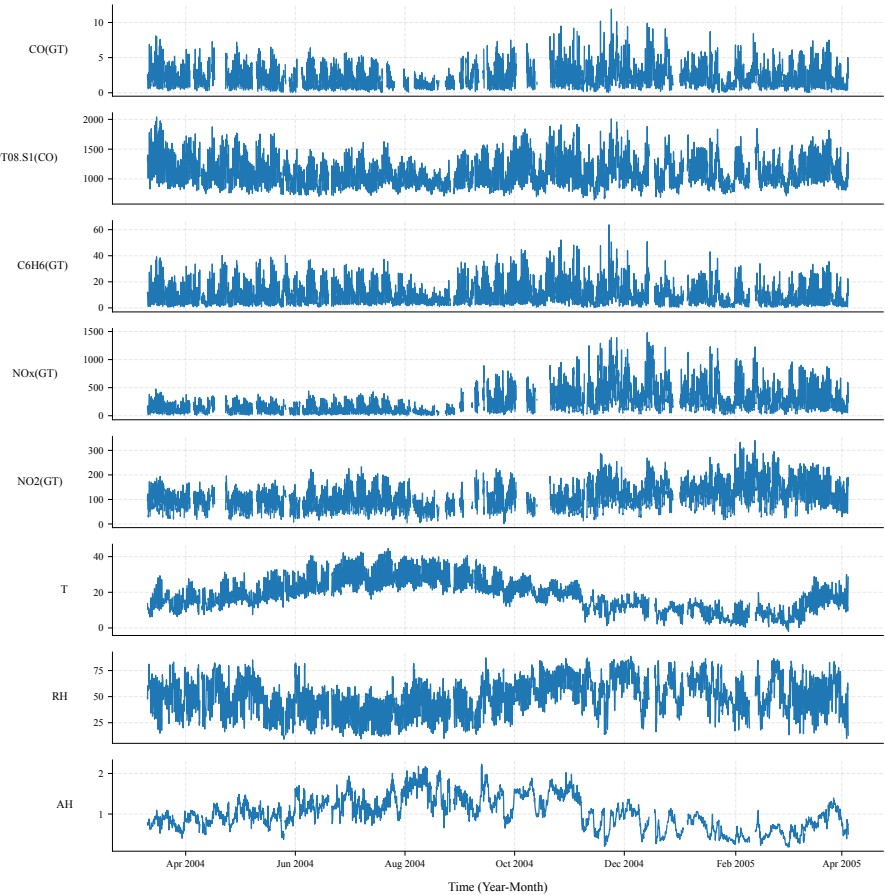

*Figure 8.* Time series of the 8 observed dimensions in the UCI Air Quality dataset. The gaps in the signals correspond to periods of sensor censorship (missing data). The framework must infer the latent state during these blackout periods without direct observation.

*Table 5.* ESS summary statistics, over all time steps.

| Experiment | $N$ | Median ESS | Min ESS | Steps below $0.1N$ |
|---|---|---|---|---|
| Linear-Gaussian | 100 | 45.6% | 16.1 | 0.0% |
| Air Quality | 20 | 54.3% | 1.0 | 0.8% |

observations, and a real world dataset, yet the median ESS still sits around 11 out of 20, i.e. above 50% of the maximum. The minimum drops to 1 at a single time step but generally stabilizes during training.

Overall these diagnostics support that the proposed importance-sampling scheme behaves robustly, even in regimes that go beyond the assumptions of our theoretical analysis.

### E.5. Ablation on the particle count $N$

Beyond the ESS diagnostics, we also evaluated how the particle count $N$ affects the algorithm's overall performance. Figure 10 plots the convergence of the ELBO proxy and the parameter estimates (transition and emission MAE) over time for the 2D linear Gaussian experiment, comparing $N \in \{10, 20, 500\}$.

The empirical results align with theoretical expectations: increasing $N$ mitigates the $\mathcal{O}(1/N)$ bias inherent in the self-normalized importance weights, which in turn accelerates ELBO convergence and reduces the asymptotic error for all parameters. As expected, the configuration with $N = 500$ yields the fastest convergence and the tightest confidence intervals. However, it is noteworthy that even in the low-sample regime ($N = 20$, corresponding to the setting used in our Air

**(a) Linear-Gaussian (N=100)**    **(b) Air Quality (N=20)**

ESS ——— Median ·····    Max (100%) - - -    Collapse (10%) - - -

*Figure 9.* Effective sample size (as % of $N$) over time on the linear-Gaussian SSM (left, $N = 100$) and the Air Quality data (right, $N = 20$). Dotted gray: pooled median. Green dashed: maximum. Red dashed: $10\%$ collapse threshold.

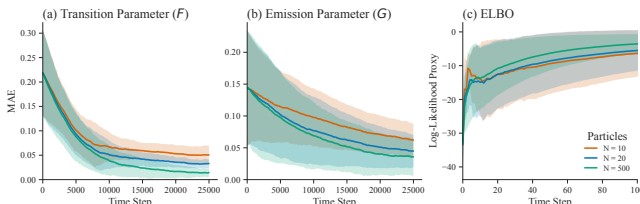

*Figure 10.* Ablation study on the number of particles $N \in \{10, 20, 500\}$. (a) MAE of the transition parameter. (b) MAE of the emission parameter $G$. (c) Evolution of the ELBO proxy. Shaded regions denote the standard deviation across 5 independent runs.

Quality experiment), the algorithm remains highly stable and achieves competitive parameter estimation without diverging, highlighting its practical efficiency.

---

**Algorithm 3** One iteration of the online gradient ascent algorithm (for $t \geq 1$) in the amortized scheme

---

**Require:**

- Previous statistics $\{(\hat{h}_{t-1}^{\phi_{t-1},i}, \hat{u}_{t-1}^{\phi_{t-1},i}, \hat{v}_{t-1}^{\theta_{t-1},i})\}_{i=1}^N$, and previous samples $\{\xi_{t-1}^i\}_{i=1}^N$;

- Intermediate quantity $a_{t-1}$, parameter estimates $(\theta_t, \phi_t)$, step sizes $(\gamma_t^\theta, \gamma_t^\phi)$;

- New observation $y_t$.

**Ensure:** $\{(\hat{u}_t^{\phi_t,i}, \hat{h}_t^{\phi_t,i})\}_{i=1}^N$, $\phi_{t+1}$, $a_t$.

    Set $a_t \leftarrow \mathcal{A}^{\phi_t}(a_{t-1}, y_t)$ and $\eta_t^{\phi_t} \leftarrow f^{\phi_t}(a_t)$, the parameters of $q_t^{\phi_t}$

    Sample $\{\xi_t^i\}_{i=1}^N$ independently from $q_t^{\phi_t}$

    **for** $i \leftarrow 1$ to $N$ **do**

        Set $\tilde{\eta}_t^{\phi_t,i} \leftarrow \tilde{f}^{\phi_t}(\xi_t^i)$

        **for** $j \leftarrow 1$ to $M$ **do**

            *// Backward sampling step, $M$ is the number of backward samples*

            Sample $(j_k)_{k=1}^M \overset{\text{i.i.d.}}{\sim} \mathsf{Cat}(\{\bar{w}_{t-1|t}^{\phi_t,i,j}\}_{1 \leq j \leq N})$ with the weights (20).

        **end for**

    Set *// Recall that each term $\ell_t^{\theta,\phi_t}$ depends on $y_t$.*

$$\hat{h}_t^{\phi_t,i} \leftarrow \frac{1}{M} \sum_{k=1}^M \left( \hat{h}_{t-1}^{\phi_{t-1},j_k} + \ell_t^{\theta,\phi_t}(\xi_{t-1}^{j_k}, \xi_t^i) \right);$$

$$\hat{u}_t^{\phi_t,i} \leftarrow \frac{1}{M} \sum_{k=1}^M \left\{ \hat{u}_{t-1}^{\phi_{t-1},j_k} + \nabla_\phi \log q_{t-1|t}^{\phi_t}(\xi_{t-1}^{j_k}, \xi_t^i) \left( \hat{h}_{t-1}^{\phi_{t-1},j_k} + \ell_t^{\theta,\phi_t}(\xi_{t-1}^{j_k}, \xi_t^i) - \hat{h}_t^{\phi_t,i} \right) \right\};$$

$$\hat{v}_t^{\theta_t,i} \leftarrow \sum_{i=1}^N \bar{w}_{t-1|t}^{\phi_t,i,j} \left\{ \hat{v}_{t-1}^{\theta_{t-1},j} + \nabla_\theta \ell_t^{\theta,\phi_t}(\xi_{t-1}^j, \xi_t^i) \right\};$$

    *// Note the difference with (22) and the inclusion of control variate $\hat{h}_t^{\phi,i}$ for the computation of $\hat{u}_t^{\phi,i}$*

    *// $\nabla q_{t-1|t}^{\phi_t}(\xi_{t-1}^{j_k}, \xi_t^i)$ is typically computed with automatic differentiation*

    **end for**

    Set

$$\widehat{\nabla}_\phi \mathcal{L}_t^{\theta_t,\phi_t} \leftarrow \frac{1}{N} \sum_{i=1}^N \left\{ \hat{u}_t^{\phi_t,i} + \nabla_\phi \log q_t^{\phi_{t-1}}(\xi_t^i) \left( \hat{h}_t^{\phi_t,i} - \frac{1}{N} \sum_{k=1}^N \hat{h}_t^{\phi_t,k} \right) \right\};$$

$$\phi_{t+1} \leftarrow \phi_t + \gamma_t^\phi \left( \widehat{\nabla}_\phi \mathcal{L}_t^{\theta_t,\phi_t} - \widehat{\nabla}_\phi \mathcal{L}_{t-1}^{\theta_{t-1},\phi_{t-1}} \right)$$

$$\widehat{\nabla}_\theta \mathcal{L}_t^{\theta_t,\phi_t} \leftarrow \frac{1}{N} \sum_{i=1}^N \hat{v}_t^{\theta_t,i};$$

$$\theta_{t+1} \leftarrow \theta_t + \gamma_{t+1}^\theta \left( \widehat{\nabla}_\theta \mathcal{L}_t^{\theta_t,\phi_t} - \widehat{\nabla}_\theta \mathcal{L}_{t-1}^{\theta_{t-1},\phi_{t-1}} \right).$$

*// Note the difference with Algorithm 2 and the inclusion of the control variate $\frac{1}{N} \sum_{i=1}^N \hat{h}_t^{\phi,i}$ in the calculation of $\widehat{\nabla}\mathcal{L}_t^{\theta,\phi}$*

*// $\nabla \log q_t^{\phi_{t-1}}(\xi_t^i)$ is typically computed using automatic differentiation*

