# OpenReview forum: "Efficient Online Variational Estimation via Monte Carlo Sampling"
_ICML.cc/2026/Conference — ICML 2026 regular_

### Official Review · Reviewer_1MEC · 2026-02-22

**Soundness:** 4
**Presentation:** 2
**Significance:** 4
**Originality:** 3
**Overall Recommendation:** 5
**Confidence:** 3

**Summary:**

The paper tackles the problem of online inference for streaming data in a state space model (SSM). They extend previous work using a variational approximation corresponding to a backwards decomposition of the target. The paper proves that ELBO maximisation has a sound theoretical basis, and derives a new related practical recursive inference algorithm. The authors show this outperforms other methods in several challenging examples.

**Compliance With Llm Reviewing Policy:**

Affirmed.

**Final Justification:**

The rebuttal has addressed my minor concerns (mainly about the paper's presentation), so I've increased my score from 4 (weak accept) to 5 (accept). The other aspects of the paper already seemed strong to me, as in my original review.

**Key Questions For Authors:**

Questions 1-5 relate to the "Strengths and Weaknesses" section. Answering these would help me increase my score. Questions 6-7 are less important queries that occurred while reading the paper.

1. How would you expect the OSIWAE algorithm of Mastrototaro et al (2025) to perform on the examples?
2. Can you give more details on the right hand column of page 3 (see comments in "Weaknesses").
3. Can you give a brief summary of how restrictive Assumption 4.2 is?
4. Can you briefly explain how the algorithm deals with missing values for the air-quality example.
5. How well justified is it to use Adam updates rather than Robbins-Monroe updates?
6. How well does the importance sampling procedure work? Does it always achieve reasonable ESS in practice? Would you expect performance to decay for high dimensional states?
7. If resources are available, could you make multiple updates for each $y_t$ value?

**Limitations:**

Yes

**Strengths And Weaknesses:**

The criteria mentioned in the review form are highlighted in bold below.

## Strengths

* The paper adds more theoretical support to using a recently introduced "COLBO" method. This provides technical **soundness**, although I wasn't able to check the proofs in detail.
* The paper is nicely self-contained and explains the area, methods and theory well without sacrificing rigour. So **presentation** is good in most places - but see "weaknesses" for a few important points I didn't follow.
* Fast on-line inference is an important problem, as illustrated by some of the paper's examples. So the paper has **significance**.
* The paper proposes a new method with improved performance, providing **originality**.

## Weaknesses (main)

* The paper builds on the theoretical framework of Mastrototaro et al (2025). But as far as I can tell, it doesn't compare to their algorithm.
* I found the right hand column of page 3 hard to follow in some places. In particular, I struggled to follow the sentence starting "Eqn. (6) ensures..." due to the following questions:
  - "$q_t^\phi$ is chosen as a parametric distribution... defined by some parameter $\eta_t$". How does it depend on $\eta_t$?
  - How is $\tilde{f}^\phi(x_t)$ defined? Is its codomain also $\mathcal{E}$?
  - How is $\eta_{t-1}^\phi$ defined? Is it the same as $\eta_{t-1}$?
* The validity of Assumption 4.2 is only discussed in an appendix that is quite technical.
* In Section 6.3, I didn't understand how the algorithm is applied to data with missing observations.

## Weaknesses (minor)

* Page 2, line 95. What do the blackboard style brackets mean?
* The algorithms talk about Robbins-Monroe updates, but the appendix says Adam updates are used.

---

> ### Author Rebuttal · Authors · 2026-03-31
>
> We thank the reviewer for highlighting the novelty of our approach, the quality of our theoretical and technical results, and their insightful questions and remarks. We address the concerns below:
>
> * **Comparison with OSIWAE:** OSIWAE relies on SMC, where sequential particle resampling can lead to path degeneracy and limits GPU parallelization when estimating the smoothing posterior. Furthermore, its IWAE objective degrades the gradient signal-to-noise ratio (SNR) for the variational parameters (Rainforth et al., 2018). In contrast, our approach uses i.i.d. samples rather than resampled particles. By explicitly modeling the backward kernels (Eq. 6), we obtain independent, degeneracy-free trajectories, while our score-function estimator preserves a high SNR. A preliminary linear Gaussian experiment (1,000 samples) confirms that our method converges in roughly one-quarter of the time with lower error (see anonymous link: https://postimg.cc/jWSn1d5z).
>
> | Method | Transition $F$ (MAE) | Emission $G$ (MAE) |
> | :--- | :--- | :--- |
> | **RMCVI (Ours)** | **0.0223** | **0.0391** |
> | OSIWAE | 0.0280 | 0.0476 |
>
> *(Steady-state MAE computed over the last 10,000 steps.)*
>
> * **Notation details on page 3:** There were some inconsistencies in the text regarding the presence of $\phi$ in the superscript for certain $\eta$ terms. In line with the remark of Reviewer umyC (Weakness 1), we have removed the dependence on $\phi$ for all natural parameters to improve clarity. Specifically:
>     * Each $q\_t$ belongs to the exponential family, with $\eta\_t$ as its natural parameter. Thus, $q\_t$ is fully determined by $\eta\_t$, and a recursion over $\eta\_t$ is sufficient to define the sequence of $q\_t$.
>     * In our experiments, $\tilde{f}^\phi(x\_t)$ is a neural network that takes $x\_t$ as input and outputs a natural parameter in $\mathcal{E}$. The parameter $\phi$ denotes the network's weights and biases.
>     * Terms like $\eta\_{t-1}^\phi$ were intended to be simply $\eta\_{t-1}$. As mentioned above, all $\phi$ superscripts on natural parameters have now been removed. $\eta\_{t-1}$ is simply a natural parameter output by the neural network parameterized by $\phi$.
>
> * **Assumption 4.2** requires that the chain $(X\_t, Y\_t, a\_t)\_{t \in \mathbb{N}}$ be uniformly ergodic, which is a minimal condition for establishing the existence of a limiting distribution for the extended chain $Z$ defined in Section 4, at least using the techniques known to us. In Appendix C.2, we identify specific—and realistically interpretable—conditions on the data-generating process and the recursively generated, user-defined states $(S\_t)\_{t \in \mathbb{N}}$ under which such a geometric contraction can be shown to hold, if not for all bounded measurable functions, as assumed in Assumption 4.2, then at least for a subset of these functions that are Lipschitz with respect to the states. For reasons of space, this was not developed further in the original version; however, prompted by the reviewer's comment, we will add a paragraph immediately following Assumption 4.2, in which this is explained and developed with full clarity.
>
> * **Missing values for Air Quality example:** Missing observations are handled by omitting the emission term $\log p(y\_t \mid x\_t)$ from the ELBO. While the forward pass bridges these gaps using the learned transition dynamics, accurate imputation relies on the explicit backward kernels (Eq. 6). These kernels propagate future information backward, enabling direct sampling from the full smoothing posterior to reconstruct the missing intervals.
>
> * **On the use of Adam:** Our algorithm is theoretically framed as a stochastic approximation (Robbins-Monro) scheme, which provides formal guarantees in our setting. In practice, we use Adam for optimization, as it yields robust and stable performance across a wide range of applications. We note that in very low SNR regimes, the Robbins-Monro approach is theoretically preferable, since its decaying learning rate ensures that the variance vanishes over time. We clarified this practical-versus-theoretical distinction in the revised manuscript.
>
> * **Effective Sample Size (ESS):** The SNIS procedure works robustly in practice, consistently achieving a healthy ESS without path collapse (e.g., a median ESS of 10.9 for $N=20$ in the Air Quality task; see this anonymous link: https://postimg.cc/2V2xS5t0). From a theoretical standpoint, high-dimensional states do not pose structural issues for our framework. However, due to the curse of dimensionality inherent in the underlying Monte Carlo approximations, we do expect empirical performance to eventually decay in highly dimensional settings.
>
> * **Multiple inner-loop steps:** Yes, it is entirely possible to perform multiple inner-loop gradient steps per observation, which would likely improve sample efficiency if desired.

---

> > ### Author Rebuttal · Reviewer_1MEC · 2026-04-01
> >
> > Thanks for the helpful replies and running extra experiments. This has addressed all my queries, so I will increase my score from 4 (weak accept) to 5 (accept).

---

### Official Review · Reviewer_hHp5 · 2026-03-08

**Soundness:** 3
**Presentation:** 2
**Significance:** 3
**Originality:** 3
**Overall Recommendation:** 4
**Confidence:** 1

**Summary:**

This paper presents an online learning framework for variational estimation in parametric state-space models. They first propose a new variational family in a recursive manner, under which the backward kernels are hugely simplified and MC methods are easy to implement.
Under this family, they derive a Bellman-type recursion expression for the ELBO and its gradient, and reduce the calculation of maximizer to a stochastic approximation problem. Then they use the classic Robbins–Monro method to solve the approximation problem and prove an asymptotic convergence result for such method under certain technical assumptions. To practically implement this framework, they introduce an importance sampler as an approximation of expectations within ELBO estimation. At last, they test the practical performance of their algorithms on three different settings, containing both synthetic and real-world problems. Compared with baseline models, better performance are obtained through those experiments.

**Compliance With Llm Reviewing Policy:**

Affirmed.

**Final Justification:**

Originality & Significance:
This paper presents an online learning framework for variational estimation in parametric state-space models. They propose a new variational family in a recursive manner, under which the backward kernels are hugely simplified and MC methods are easy to implement.  Under this family, they derive a Bellman-type recursion expression for the ELBO and its gradient, and reduce the calculation of maximizer to a stochastic approximation problem. The Bellman-type expression for ELBO and its gradient is interesting and novel.

Soundness:
This paper is technically sound. Every claim and statement are justified appropriately. Regarding the proofs related sampling, they only provide asymptotic bounds which is far away from practical use. However, such a problem seems to be a hardness result in this field instead of a weakness of this paper.

Clarity:
As other reviewers suggested, writings can be improved by simplifying their notations and proofs.

Given the fact that the reviewer is not an expert in variational online learning, my score of weak accept is solely an educated guess.

**Key Questions For Authors:**

1. Assumption 4.4 implies that the potential functions need to be approximately uniform, which seems to be strong. Is this assumption commonly used in this field? If not, could the authors justify it either through intuitive explanations or some empirical evidence? In practice, do we obtain an estimations of the scale of $\epsilon$?
2. Is it possible for us to obtain some error control of the online Markov Carlo estimation step?
3. Is it possible for us to have a non-asymptomatic version of equation (12)? i.e. the convergence rate is a function of $t$.

**Limitations:**

Yes

**Strengths And Weaknesses:**

Strength:
1. The Bellman-type expression for ELBO and its gradient is interesting and novel.
2. This paper is technically sound. Every claim and statement are justified appropriately.

Weakness:
1. Readability could be improved by providing some intuitive explanations for key assumptions and theorems.
2. The theoretical contribution of the convergence results in equation (12) is rather limited due to: (a) Assumptions 4.2-4.4 seems to be strong and the expressive power of the variational families that satisfy those assumptions may be restricted; (b) It only gives an asymptotic convergence, which does not provide a direct explanation for the power of finite-time approximate algorithms.

---

> ### Author Rebuttal · Authors · 2026-03-31
>
> We thank the reviewer for highlighting the novelty of our approach, the quality of our theoretical and technical results, and for the insightful questions and remarks.
>
> Concerning the readability—specifically providing more intuitive explanations for the theoretical parts—we plan, if the paper is accepted, to use the additional page to expand these explanations.
>
> More specifically (**KQX** refers to Key Question X, **WX** to Weakness X):
>
> * **W2: Assumption 4.2.** Assumption 4.2 requires that the chain $(X\_t, Y\_t, a\_t)\_{t \in \mathbb{N}}$ be uniformly ergodic, which is a minimal condition for establishing the existence of a limiting distribution for the extended chain $Z$ defined in Section 4, at least using the techniques known to us. In Appendix C.2, we identify specific—and more or less realistic—conditions on the data-generating process and the recursively generated, user-defined states $(S\_t)\_{t \in \mathbb{N}}$ under which such a geometric contraction can be shown to hold. If not for all bounded measurable functions as assumed in Assumption 4.2, then at least for a subset of these functions that are Lipschitz with respect to the states. For reasons of space, this was not developed further in the original version, but prompted by the reviewer's comment, we added a paragraph immediately following Assumption 4.2, in which this is explained and developed with complete clarity.
>
> * **KQ1 and W2: Assumption 4.4.** In our algorithm, the $\psi\_t$ functions are chosen to be conjugate to the variational filtering distributions (the $q\_t$'s); in practice, these are taken to be Gaussian. Moreover, in practice, the algorithm's output can be restricted to a sufficiently large, compact subset of $\mathbb{R}^d$, ensuring—by the extreme value theorem—that the potential functions attain finite maxima and minima on this set. A similar line of reasoning is common in the literature on state-space models and particle methods, where it is typically assumed that the relevant Markov kernels render the entire state space a 1-small set, thereby guaranteeing uniform ergodicity (see, for instance, Douc, Moulines, and Stoffer, 2014, Chapter 13).
>
> * **KQ2: Error Control of MC estimation.** A classical $\mathcal{O}(1/N)$ bias analysis could be carried out to quantify the discrepancy between the theoretical update and its empirical counterpart, providing explicit finite-sample error bounds. Possible directions are as follows:
>     * A refined analysis, inspired by recent advances such as the results presented in Surendran et al. (NeurIPS 2025), could be adapted to our setting. These modern tools allow for a comprehensive non-asymptotic study of stochastic optimization with biased gradients and adaptive steps for non-convex smooth functions.
>     * It would be interesting to investigate bias-reduction strategies beyond the SNIS framework—for instance, along the lines of BR-SNIS (Cardoso et al., 2022)—which may offer alternative ways to control or mitigate bias in practical implementations.
>
>   While these directions fall outside the scope of the present paper, as our main objective is to provide a theoretically grounded framework for online variational learning, which is already a technically dense and challenging task, they highlight promising avenues for future work. In particular, a non-asymptotic analysis of the proposed algorithm constitutes, in our view, an interesting and valuable research direction. A discussion of these points has been added to the revised version.
>
> * **KQ3: Non-asymptotic version of Equation (12).** We have not established such a result for this specific context. However, a natural lead to explore would be the theory of Markov chains, and specifically concentration inequalities such as the Azuma-Hoeffding bounds derived in Douc et al. (*The Annals of Statistics*, 2011).

---

> > ### Author Rebuttal · Reviewer_hHp5 · 2026-03-31
> >
> > Thanks for the detailed response.
> >
> > I admit providing rigorous gurantees for MC estimation is beyond this paper's scope. Given the fact that I am not an expert in  online variational learning, and my research expertise mainly foucses on sampling theorey, I could only come up with questions related to that. Assuming ergodicity of Markov chain is indeed a mild assumption, but $\alpha$ could be notoriously bad and lead to an exponential mixing time. I suppose such a problem should be viewed as a hardness result of the field instead of the weakness of this paper.
> >
> > Since I am not an expert in this area, I shall keep the score and confidence and leave the discussion to other expert reviewers.

---

### Official Review · Reviewer_sDU1 · 2026-03-12

**Soundness:** 3
**Presentation:** 2
**Significance:** 3
**Originality:** 2
**Overall Recommendation:** 4
**Confidence:** 4

**Summary:**

This paper proposes RMCVI, an online variational inference algorithm for SSMs that jointly learns model and variational parameters. The key idea is replacing the expensive regression-based inner optimization of Campbell et al. (2021) with self-normalized importance sampling, exploiting a link between variational filtering and backward kernel distributions. The method comes with a (partial) theoretical justification, and demonstrates ~5x speedups over the main baseline with comparable accuracy.

**Compliance With Llm Reviewing Policy:**

Affirmed.

**Final Justification:**

The authors have addressed my main concerns

**Key Questions For Authors:**

1. How does the effective sample size behave during runs?
2. How much bias accumulates in long recursive runs?
3. How tight is the surrogate COLBO objective?

**Limitations:**

The limitations of the proposed method are not adequately discussed

**Strengths And Weaknesses:**

# Strengths
1. Frames online variational EM as a Robbins-Monro stochastic approximation with a provable limiting objective.
2. Replaces regression steps with a importance sampling approach derived naturally from the variational family, avoiding backward messages and complex inner optimization loops.
3. Achieves ~5x speedups over prior regression-based variational smoothers while maintaining comparable estimation performance.
4. Experiments include challenging scenarios (e.g., nonlinear dynamics, missing data, sensor failures in air quality benchmarks) and show that the method remains robust.

# Weaknesses
1. The theoretical results analyze an idealized setting (e.g., fixed parameters and bounded potentials), while the implemented algorithm uses online parameter updates and SNIS approximations. The relationship between the theory and the actual algorithm is not clearly analyzed.
2. The ergodicity assumptions required for the theory (e.g., uniform ergodicity of the augmented chain) are not discussed or verified for the nonlinear or chaotic models used in experiments.
3. The algorithm relies on self-normalized importance sampling (SNIS), which introduces O(1/N) bias that may accumulate in long recursive runs, but the paper does not analyze this effect.
4. With relatively small particle counts (e.g., N=20 in the air quality experiment), importance weight degeneracy could be a concern, yet no effective sample size (ESS) diagnostics or discussion of failure modes are provided.
5. Important stability components, such as variance-reduction techniques (e.g., control variates), are relegated to the appendix rather than explained in the main algorithm description.
6. Comparisons are limited to a small number of baselines, primarily the method of regression-based VI for SSMs. Other relevant approaches are not included or discussed, such as PaRIS, Variational Recurrent Neural Network, or recent online variational inference work like Online Variational Inference for State Space Models.
7. The method optimizes a surrogate COLBO objective, but the paper does not analyze how tight this bound is relative to the true objective or how the gap behaves in the online setting.
8. There are no ablations studying how performance depends on the number of samples N, which makes it difficult to assess estimator variance or stability.
9. The impact of the parameter lag used in the updates is not empirically analyzed or theoretically justified.
10. Section 4 and the appendix are notation-heavy and difficult to follow, with limited bridging explanation between theoretical expressions and their implementation.
11. Key implementation details such as hyperparameters and learning-rate choices are not clearly documented, which makes reproducibility harder.

---

> ### Author Rebuttal · Authors · 2026-03-31
>
> We thank the reviewer for highlighting the strengths of our method. We address the concerns below, grouping related points together (**KQX** refers to Key Question X, **WX** to Weakness X):
>
> * **W1, W2, W3, KQ2: Relation between theoretical and practical algorithm.** The main objective of the paper is to provide a theoretically grounded framework for online variational learning, which is already a challenging task. To highlight the practical relevance of our results for the ICML community, we also design a computationally efficient online estimator of the COLBO and its gradient. This estimator introduces some approximations that are not covered by our theoretical analysis.
> A key difficulty in analyzing the practical implementation arises from the use of SNIS for the recursive gradients, as it introduces an $\mathcal{O}(1/N)$ bias.
> We believe future avenues to study or reduce this bias could be:
>     * A classical $\mathcal{O}(1/N)$ bias analysis could be carried out to quantify the discrepancy between the theoretical update and its empirical counterpart, providing explicit finite-sample error bounds.
>     * A refined analysis, inspired by recent advances such as the results in Surendran et al. (NeurIPS 2025) that provide modern tools for studying non-asymptotic behavior, could be adapted to our setting. This approach enables a comprehensive non-asymptotic analysis of stochastic optimization with biased gradients and adaptive steps for non-convex smooth functions.
>     * An analysis of the impact of bias reduction strategies beyond the SNIS framework, along the lines of Cardoso et al. (NeurIPS 2022), could provide alternative ways to control or mitigate bias in practical implementations.
>
>     A discussion has been added in the revised version. However, for the specific point raised in **KQ2**, there is no reason why the bias should accumulate across iterations.
>
> * **W4, W8, W11, KQ1: ESS and number of samples $N$.** The ESS remains stable and avoids weight collapse across all experiments. Even in our Air Quality experiment with only $N = 20$ samples, the ESS stays well above typical collapse thresholds (e.g., a median ESS of 10.9 for $N=20$), as shown in the following anonymous link: https://postimg.cc/2V2xS5t0. To evaluate the impact of the particle count, we varied $N \in \{10, 20, 500\}$, the results of which are shown in the following plot: https://postimg.cc/GB5k5cMK.
> These empirical results align with theoretical expectations: increasing $N$ accelerates ELBO convergence and reduces MAE for all parameters, showing the predictable scaling of our method.
> Both the ESS analysis and the particle-count study have been added to the appendix of the revised version. Additionally, we include a detailed list of hyperparameters to ensure reproducibility.
>
> * **W7, KQ3: Tightness of the COLBO.** Since the exact COLBO (Eq. 7) is a time-normalized limit of the ELBO (Eq. 4), it has, in principle, the same tightness properties as the ELBO. In particular, the gap is given by the limiting KL divergence between the variational distribution and the true joint-smoothing distribution. Hence, its tightness is governed by the expressiveness of the variational family, as in standard variational inference. Interestingly, the objective admits a natural extension, analogous to the *importance-weighted autoencoder* (IWAE), in which $M$ multiple samples from the variational distribution are used to form a tighter bound. This would yield a family of increasingly tight objectives that approach the true likelihood as $M$ increases, and we believe it would be possible to show theoretically that the gap is $\mathcal{O}(1/M)$ for such an approach. We leave a full development of this extension to future work, but have included a discussion in the revised version. As for the MC approximation of the COLBO in Section 5, we expect it to have an $\mathcal{O}(1/N)$ bias relative to the exact COLBO (and, consequently, an $\mathcal{O}(1/M + 1/N)$ relative to the likelihood for the IWAE extension).
>
> * **W6: Other Baselines.** We added comparisons with other online variational learning methods: OVSMC and OSIWAE (see our response to Reviewer umyC and 1MEC), which demonstrate the performance of our algorithm. Our experimental design deliberately focused on comparisons with online variational inference schemes.
> The PaRIS algorithm is cited in the related work (Paragraph "SMC for online learning in SSMs"). Implementing PaRIS would require integrating it into a learning algorithm, such as online EM (Olsson and Westerborn, IFAC-PapersOnLine, 2015) or a recursive MLE based on a tangent filter approximation (Olsson, Westerborn and Alenlöv, AISM, 2020). As reported in the original papers, PaRIS-based recursive MLE exhibits slower convergence compared to OVSMC and OSIWAE. Due to time constraints, we were unable to implement it; however, we could certainly include it in a final version if desired.

---

> > ### Author Rebuttal · Reviewer_sDU1 · 2026-04-04
> >
> > I appreciate the effort in providing the ESS diagnostics, the ablation on $N$, and the comparisons with OVSMC and OSIWAE. These results address some of my concerns.
> >
> > *The Bias Accumulation Concern (W1, W3, KQ2)*
> > In recursive online estimation, even a small per-step bias can compound into a significant parameter drift over long sequences. Since the paper's primary strength is its theoretical grounding, I'd like to see a more rigorous discussion on this gap between the "unbiased theory" and the "biased implementation" about the stability of these updates in the long run.
> >
> > *COLBO Tightness (W7, KQ3)*
> > Noted on the theoretical argument that COLBO has the same properties as ELBO. However, in an online setting with parameter lags, things can get messy. It would be great if the authors could include a quick experiment or a more targeted discussion in the appendix comparing the COLBO values against a ground-truth likelihood for a toy model. This would actually prove the "same tightness" claim holds up in practice.
> >
> > ---
> >
> > Edit: The authors have addressed my concerns, I have therefore increased my score.

---

> > > ### Author Response · Authors · 2026-04-06
> > >
> > > **Bias Accumulation Concern**
> > >
> > > We agree with Reviewer sDU1 that in recursive online estimation, even a small per-step bias can accumulate and significantly impact parameter estimates. If the reviewer is referring to the bias introduced by the practical algorithm using Monte Carlo estimates, we agree that this is an important concern. At the same time, recent literature provides several useful insights into this issue.
> > >
> > > - First, as detailed in the rebuttal, a classical $O(1/N)$ bias analysis could be carried out by adapting the analysis of *Olsson and Westerborn (2017), Efficient particle-based online smoothing in general hidden Markov models: the PaRIS algorithm, Bernoulli* to our Monte Carlo–based updates. This would allow us to control the discrepancy between the theoretical update and its empirical counterpart, thereby providing explicit finite-sample error bounds. In addition, it is known that this bias can be reduced using approaches such as those proposed in *Cardoso et al. (2022), BR-SNIS: Bias Reduced Self-Normalized Importance Sampling, NeurIPS*. This suggests that the bias at each iteration can be controlled with an explicit dependence on $N$, meaning that the Monte Carlo sample size can be tuned across iterations to satisfy suitable conditions for convergence.
> > >
> > > - Second, because we formulate our variational approach within a stochastic approximation framework, we can leverage recent results such as *Karimi et al. (2019), Non-asymptotic Analysis of Biased Stochastic Approximation Schemes, COLT*, as well as more recent extensions (e.g., *Surendran et al. (2024), Non-asymptotic Analysis of Biased Adaptive Stochastic Approximation, NeurIPS*). These results make it possible to establish convergence to a critical point of the objective, with rates that explicitly depend on the bias at each iteration. This, of course, requires assumptions on the objective function, such as the Polyak--Łojasiewicz condition, $L$-smoothness, and assumptions on the mean-field bias, which in our setting is expected to scale as $1/N$ (as discussed above).
> > >
> > > To conclude, we agree that this bias topic is of great importance with many practical implications, and we have added a dedicated discussion to clarify the position of our paper. Given the already technical nature of our contribution, we believe that a more detailed treatment would be better suited to a separate work focusing on non-asymptotic bounds that could guide the practical tuning of hyperparameters ($N$, $K$ for IWAE, and bias reduction techniques).
> > >
> > > **COLBO Tightness**
> > >
> > > We now empirically assess the tightness of the bound provided by the online COLBO on a 2D linear Gaussian state-space model (SSM), simulated with a true parameter $\theta^*$, for which the average log-likelihood ($\lambda(\theta)$ in the paper) can be computed exactly. This evaluation is performed on four distinct datasets, each with five independent random initializations. The anonymous plot: https://postimg.cc/DJvhzS40 shows the results.
> > >
> > >   - The blue line shows the approximated COLBO $\frac{1}{t}\mathcal{L}^{\hat{\theta}_t,\hat{\phi}_t}_t$ over time, where $\hat{\theta}_t$ and $\hat{\phi}_t$ are updated at each step using our algorithm.
> > >   - The green line shows $\frac{1}{t}\log(p^{\hat{\theta}{t}}(Y_{1:t}))$.
> > >   - The red line shows $\frac{1}{t}\log(p^{\theta^*}(Y_{1:t}))$.
> > >
> > >   We observe (for instance, in the top-left panel) that the learned parameters yield a log-likelihood close to that of the true parameter, and that the gap between our contrast and that of a true recursive maximum likelihood setting decreases significantly as the posterior distribution is learned (thanks to the updates of $\hat{\phi}_t$).
> > >
> > >   This illustration shows empirical proof of the tightness. We acknowledge that it is based on a toy example where the variational family contains the true posterior distribution. For more complex cases, recent results (*Chagneux et al. (2024), Additive smoothing error in backward variational inference for general state-space models, JMRL*) show that the error in additive smoothing (in our example, between the ELBO and the log-likelihood) grows, in ideal variational settings, at most linearly with $t$ (see Remark 4 of the cited paper). As our contrast is the ELBO averaged over time, this result suggest that the error is well controlled over time and then that online variational approaches (not only our algorithm) can achieve reasonably tight bounds.

---

### Official Review · Reviewer_umyC · 2026-03-12

**Soundness:** 3
**Presentation:** 2
**Significance:** 3
**Originality:** 4
**Overall Recommendation:** 5
**Confidence:** 3

**Summary:**

This paper tackles the challenge of learning to filter a state space model online. The model and proposal parameters are jointly optimised by maximising the COBOL, an asymptotic time analogue of the ELBO. The authors propose a simple strategy to mimic the backwards decomposition of the posterior with a chain of distributions that can be sampled forward, then reweighted to give the backward kernel. The forwardly sampled distributions are specified to follow a given natural exponential family and the reweighting factor so that the backward kernel follows the same family ensuring that the backwards kernel is a valid distribution. Theoretical results are proved about the resulting algorithm. In practice a modified version is used that approximates the ELBO and its gradients with previously computed values rather than passing through the complete computation graph each time the parameters are changed.

**Compliance With Llm Reviewing Policy:**

Affirmed.

**Ethical Review Concerns:**

None.

**Final Justification:**

Thank you authors, for your paper and rebuttal. My main isues with the paper were with it's clarity, however the authors have expressed comittment to resolving these issues. My remaining concerns were addressed in the rebuttal. This work constitues a neat (and significant) iteration on OVSMC. I believe this work could have widespread practical utility in situations where it is not feasible to collect a large amount of training data prior to commencing filtering. The authors additionally provide theoretical justification for their approach. Whilst the algorithm that is studied theoretically is substantially different from the one that is suggested for practical use, I agree with the authors that proving analogous results for the practical version would likely go far beyond a reasonable scope for an ICML paper. Therefore, I raise my score from 4 to 5 and recomend the paper for acceptance to the conference proceedings.

**Key Questions For Authors:**

1. To obtain estimates of the posterior mean, are you applying an importance sampler that weights some online approximation of the target by Equation (5), or are you assuming that the posterior variational family includes the true posterior and simply taking the proposed particles as samples from the posterior?

**Limitations:**

Yes

**Strengths And Weaknesses:**

Strengths:

1. The main methodological insight, to develop a backwards proposal kernel that can be sampled forwards by importance sampling is not something I’ve seen before and fits naturally with a powerful LSTM parameterisation. This is a simple idea that extends well established techniques in variational inference in deep state space models to a fully online setting.

2. The proposed algorithm is substantially cheaper, computationally, than its competitors.

3. Unfortunately, the proposed algorithm has is restricted to REINFORCE family gradient estimators rather than the generally preferred reparameterised estimators. However, the authors extend their work to develop a control variate stabilisation.

4. The experiments are largely comprehensive, there are not many baselines approaches tested, however the problem does not have many prior approaches in the literature.

5. Theoretical assurances are proved about a closely related, idealised, version of the proposed algorithm justifying the approach.

Weaknesses:

1. The notation is overwhelming, the explicit notation of the parameter dependence in the superscripts adds clutter, I see in some cases the time-index of the parameter superscript carries meaning, but it would be better if this could be notated a little more cleanly. Sometimes the parameters have a time-index sometimes they do not, this is confusing. The equations (13) to (16) are not endowed with such subscripts when (20-23) are; here the time-index alters how I interpret the equations. I would avoid excessively defining symbols, for example $u_{t}, v_{t}$ have a simple definition that can be kept explicit. The symbol $\ell$ is overloaded. The function osc() is not defined when used, this is not standard enough notation to be left undefined, I have not seen it previously.

2. I would prefer that all equations in the main text are numbered.

3. Algorithm 2 is formatted poorly with inconsistent capitalisation, no line numbering and inconsistent notation of a for loop. I would also either add extra vertical space between the first 4 lines or split each statement over multiple lines.

4. A lot of space is devoted to proving theoretical results about an idealised version of the algorithm that differs substantially from the practical version. I understand that such results are either hard to establish or don’t hold for the practical case, but the authors could do more to reassure the reader that the practical conditions are not too far from the idealised case for the established results to lose their relevance.

5. I would like to see comparison to OVSMC on all experiments.

6. Unlike OVSMC the proposal distributions are restricted to belong to a pre-specified natural exponential family.

---

> ### Author Rebuttal · Authors · 2026-03-31
>
> We thank the reviewer for their positive and insightful assessment of our work, in particular for recognizing the novelty and practicality of our proposed approach, as well as its computational efficiency and supporting theoretical analysis. We address the concerns below,
>
> ### Weaknesses
>
> 1. We thank the reviewer for this careful reading. We agree that the notation in some places is heavy, and we have simplified it where possible. Following this comment, we carefully revised the manuscript to clarify the notation and remove ambiguities and typos.
> In particular, as a response to the remarks of Reviewer 1MEC (Weakness 2), we have suppressed the explicit dependence of natural parameters $\eta$ on $\phi$ and $x_t$. However, some time-indexed superscripts are unavoidable; since our algorithm propagates the parameters recursively as new data arrives, certain quantities are evaluated using $\phi_t$ while others use $\phi_{t-1}$. Omitting these indices would make the distinction ambiguous and could lead to misinterpretation of recursive updates. Finally, in the revised version, we defined the oscillation seminorm $\mathsf{osc}(\cdot)$ when it is first introduced and ensured that any overloaded symbols, such as $\ell$, are clarified.
>
> 2. We respectfully note that in our paper, we number only the equations that are explicitly referred to in the text, while leaving others unnumbered. This is a common practice in most machine learning venues, including NeurIPS, ICML, and ICLR, and helps reduce clutter while keeping references clear.
>
> 3. While the `for` loop notation in Algorithm 2 follows the standard conventions of the `algorithm2e` package, we improved the overall readability by ensuring consistent capitalization, adding line numbers, and introducing extra vertical space where needed. These changes make the algorithm easier to follow without altering its structure.
>
> 4. We agree with the reviewer regarding the gap between the theoretical analysis and the practical algorithm. The objective of the paper is to provide a theoretically grounded framework for online variational learning. This allows us to introduce a computationally efficient online estimator of the COLBO and its gradient. Developing this theoretical foundation is already a substantial challenge, and we wanted to emphasize the practical relevance of the approach as it is of interest to the ICML community.
> The main difficulty in analyzing the practical implementation arises from using SNIS for the recursive gradients. While standard stochastic approximation proofs assume idealized, unbiased gradients, SNIS is consistent but introduces an $\mathcal{O}(1/N)$ bias. However, we believe that there are several interesting avenues for analyzing this bias in our setting, as well as for developing bias-reduction strategies. This will be elaborated in a dedicated paragraph, but we consider a full treatment to be beyond the scope of the paper, which is already technically demanding.
>
> 5. We included OVSMC baselines for all experiments in the revision. While OVSMC's particle-based variational distributions offer non-parametric flexibility, our restriction to the natural exponential family is a deliberate design choice that enables explicit modeling of the backward kernels (Eq. 6). In contrast, OVSMC relies on sequential particle resampling, which can cause path degeneracy, whereas our approach produces independent trajectories with high signal-to-noise gradient estimates. Preliminary results confirming this advantage can be found in the following anonymous link: https://postimg.cc/gn8H3DnW.
>
> In summary, the results are:
>
> | Method | Transition $F$ (MAE) | Emission $G$ (MAE) |
> | :--- | :--- | :--- |
> | **RMCVI (Ours)** | **0.0223** | **0.0391** |
> | OVSMC | 0.0377 | 0.0552 |
>
> *(Steady-state MAE computed over the last 10,000 steps.)*
>
> 6. We agree that this variational framework requires restricted attention to a chosen variational family; however, this is a standard feature of all variational frameworks in practice.
>
> ### Key Questions for Authors
>
> The posterior smoothing distribution (and all expectations related to it) are approximated (after training) by sampling from the variational distribution given by Equation (5), which is straightforward, as each kernel is chosen to be Gaussian.

---

> > ### Author Rebuttal · Reviewer_umyC · 2026-04-01
> >
> > Thank you for your response to my questions.
> >
> > I am happy that my concerns have largely been addressed. The authors appear commited to resolving the issues with respect to clarity and that OVSMC was included as a baseline for the outstanding experiments. I agree that further expanding the theoretical framework towards the practical approach would significantly widen the scope, beyond what can reasonably be expected for a methodological conference paper. But I am pleased that the authors plan to explicitly discuss this aspect of the work in the revision.

---

### Decision · Program_Chairs · 2026-04-30

**Decision:**

Accept (regular)

**Comment:**

This paper proposes an online algorithm for state-space models, where the smoothing distribution and a variational approximation are jointly learned as data arrive. Campbell et al. proposed a method involving a regression estimation in the inner-loop, which this paper replaces with a self-normalized importance sampling estimate. This is intuitively reasonable and seems to work well in a handful of simulations.